# The Anglo-Saxon migration and the formation of the early English gene pool

The history of the British Isles and Ireland is characterized by multiple periods of major cultural change, including the influential transformation after the end of Roman rule, which precipitated shifts in language, settlement patterns and material culture[1]. The extent to which migration from continental Europe mediated these transitions is a matter of long-standing debate[2-4]. Here we study genome-wide ancient DNA from 460 medieval northwestern Europeans—including 278 individuals from England—alongside archaeological data, to infer contemporary population dynamics. We identify a substantial increase of continental northern European ancestry in early medieval England, which is closely related to the early medieval and present-day inhabitants of Germany and Denmark, implying large-scale substantial migration across the North Sea into Britain during the Early Middle Ages. As a result, the individuals who we analysed from eastern England derived up to 76% of their ancestry from the continental North Sea zone, albeit with substantial regional variation and heterogeneity within sites. We show that women with immigrant ancestry were more often furnished with grave goods than women with local ancestry, whereas men with weapons were as likely not to be of immigrant ancestry. A comparison with present-day Britain indicates that subsequent demographic events reduced the fraction of continental northern European ancestry while introducing further ancestry components into the English gene pool, including substantial southwestern European ancestry most closely related to that seen in Iron Age France[5,6].

The first millennium CE saw major demographic, cultural and political change in Europe, including the rise and fall of the Roman Empire, migration and the emergence of medieval institutions that shaped the modern world. The post-Roman transformation of lowland Britain was particularly profound. The end of the Roman administration in fifth century Britain preceded a dramatic shift in material culture, architecture, manufacturing and agricultural practice, and was accompanied by language change[1]. The archaeological record and place names indicate shared cultural features across the North Sea zone, in particular, along the east and southeast coasts of present-day England, Schleswig-Holstein and Lower Saxony (Germany), Frisia (Netherlands) and the Jutland peninsula (Denmark)[2-4]. Examples include the appearance of *Grubenhäuser* (sunken feature buildings), large cremation cemeteries and the styles of cremation urns or objects that used animal art and chip-carved metal[7-11]. Moreover, wrist clasps, as well as cruciform and square-headed brooches, found in sixth and seventh century Britain had attested southern Scandinavian origins[12,13]. Despite these similarities across the North Sea zone, there was also insular material culture that had no continental equivalent[14,15]. Adding to this, some places and geographical features such as rivers retained names of Celtic or late Latin origin[16,17].

From the Renaissance to the present day, the primary explanatory narrative for these changes has been invasion and conquest followed by resettlement from the continent[18]. On the basis of a small set of written sources, it was supposed that the local Romano-British population was largely replaced by migrants from the Germanic-speaking part of the continent. However, the extent to which these traditional cultural historical interpretations explain patterns of material culture or agree with the historical accounts has been questioned[18]. For example, historical sources going back to Bede (writing in the eight century) indicated Jutes as settlers in Kent. But, in an issue that became known as 'the problem of the Jutes'[19-21], this historically attested migration is difficult to determine from or reconcile with the archaeological record. Indeed, material culture elements found in Kent resemble those of contemporary Merovingian France and Alemannic (southern) Germany, rather than the rest of England or Denmark. Such discrepancies between the archaeological record and historical narratives could be argued to support a rejection of migration or invasion hypotheses, and this was the preferred theoretical position of many archaeologists from the 1960s onwards[1,18,22]. By that time, many scholars favoured a model of elite dominance involving small, mobile warbands and the acculturation of the local British population. However, the available isotopic and DNA evidence, even if hitherto small scale, suggests that immigrants were less wealthy and buried alongside locals[23-28], which does not fit a model of elite influence that could explain the adoption of a West Germanic language with apparently minimal influence from Celtic or Latin[29-32].

There is a history of addressing these questions using genetic data. After early attempts to use ancient genetic data failed[33], researchers turned to studies based on present-day populations and uniparentally inherited markers, but still without reaching consensus. Work based on present-day Y chromosomes inferred 50–100% replacement of male lineages during the Early Middle Ages in eastern England[34,35]. More recently, the first genome-wide study of present-day British people

concluded that immigrant continental northern European ancestry makes up less than 50% of the present-day southeastern English gene pool[36]. However, populations change over time through drift and gene flow, so present-day populations may be poor proxies for ancient groups of unknown genetic makeup. The feasibility of ancient DNA analyses to inform on population history in Britain was first demonstrated with the report of genome-wide ancient DNA (aDNA) data[26,37] from 20 individuals from the Iron Age to the Early Middle Ages, two studies that have provided unambiguous evidence for continental ancestry in early and middle Anglo-Saxon contexts.

Here we investigate early medieval population dynamics in England and across the North Sea zone with the first large-scale genome-wide study of aDNA in this time period and region, increasing the archaeogenetic record in England specifically, from 8 to 285 individuals. We target a comprehensive time transect of sites in the south and east of England, spanning predominantly the time period 450–850 CE, starting with early Anglo-Saxon cemeteries including Apple Down, Dover Buckland, Eastry, Ely, Hatherdene Close, Lakenheath, Oakington, Polhill and West Heslerton. This allows us to address questions concerning the extent of continental migration to England, and its effect on the local insular gene pool. In addition, the association of artefacts with individuals allows us to study the dynamics of the migration process at the community level.

## New aDNA data

We sampled skeletal remains from 494 ancient northwestern Europeans from 37 different sites in England, Ireland, the Netherlands, Germany and Denmark, dated between approximately 200 and 1300 CE (Supplementary Note 1 and Supplementary Table 1). We prepared powder from skeletal material, extracted aDNA and converted it into double-stranded or single-stranded libraries (Methods). We selected 439 libraries for hybridization DNA capture to enrich for sequences that overlapped 1.24 million single-nucleotide polymorphisms (SNPs). For 40 libraries, we generated complete genomes without capture, with a mean coverage of 0.9×.

After quality filtering (Methods) and exclusion of duplicate individuals, genome-wide data for 460 individuals were available for analysis. These include 278 ancient individuals from England, and 182 individuals from neighbouring ancient populations in Ireland and the European continent (Fig. 1). We combined our newly reported data with published aDNA from 4,336 individuals (Supplementary Note 2), including 1,098 post-Neolithic genomes from northwestern Europe[26,37–44]. We also compiled a reference dataset of 10,176 present-day European individuals[36,45–47] genotyped on an intersection of 445,171 SNPs (Supplementary Note 2). To aid interpretation of our genetic data, we also radiocarbon-dated 57 samples selected on the basis of ancestry composition, burial assemblage and preservation.

## Population shifts in post-Roman England

We performed principal component analysis (PCA) on 5,365 present-day northwestern Europeans, from Ireland to Sweden, and projected our ancient genomes onto this genetic variation (Fig. 2). For present-day variation, PC1 and PC2 broadly reflect geography, forming a V-shaped pattern from Scandinavians via individuals from northern Germany and the Netherlands towards those from Britain and Ireland. We highlight the position of individuals from present-day England (Fig. 2a), which follow a clinal distribution defined by the western British and Irish (WBI; which includes Irish, Northern Irish, Scottish and Welsh) at one extreme and overlapping present-day Dutch at the other extreme. The ancient genomes fall onto a slightly separate cline, with most of the early medieval individuals from Dutch, German and Danish sites plotting on top of present-day continental northern Europeans (CNEs; northern Germans and Danish), whereas Bronze and Iron Age individuals from Britain and

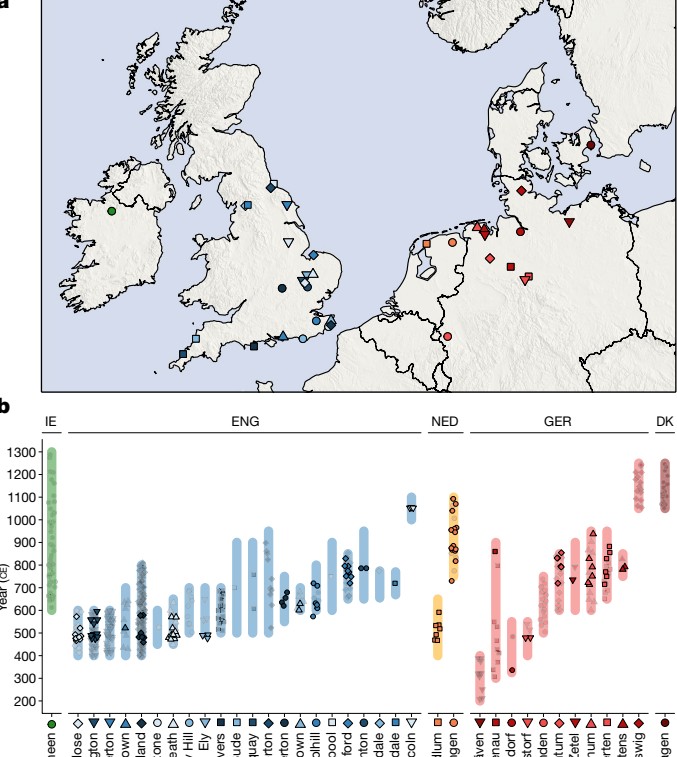

**Fig. 1 | Spatial and temporal origin of ancient individuals in this study.** **a**, Spatial distribution of sites analysed in this study. **b**, Temporal distribution of samples analysed in this study, with site occupancy ranges indicated by bars. Non-transparent symbols indicate radiocarbon-dated samples; transparent symbols are scattered uniformly along site occupancy ranges. DK, Denmark; ENG, England; GER, Germany, IE, Ireland; NED, Netherlands.

Ireland cluster together with WBI (Fig. 2b). Of note, in contrast to the preceding Bronze and Iron Age individuals from Britain and Ireland, the majority of the early medieval samples from England (England EMA) plot together with the ancient individuals from the continental North Sea area along with the present-day CNEs. The divergence between prehistoric and early medieval individuals from England is also seen in the distribution of genetic distances ($F_{ST}$) as well as shared alleles ($F_4$) on both the population (Extended Data Fig. 1) and the individual scale (Supplementary Fig. 3.3). We notice that the individuals from early medieval English sites are distinctly heterogeneous in the first two PCs and cover the full extent of the cline between the Bronze and Iron Age cluster and the early medieval cluster.

These genetic patterns suggest that early medieval individuals from England have variable amounts of CNE ancestry. Although most individuals from early medieval English sites cluster clearly with either present-day WBI samples or CNEs, many individuals fall between these two clusters, suggesting admixture between these ancestral groups. To quantitatively estimate these ancestry compositions, we decomposed ancestral sources using a supervised clustering approach implemented in the software ADMIXTURE[48]. Specifically, we assembled modern populations into two metapopulations that serve as proxies for the source ancestries in early medieval England defined above: CNE ($n = 407$) and WBI ($n = 667$). We confirmed that these two present-day metapopulations accurately represent the ancient admixture sources by testing their relationships to the ancient individuals from England using $F_{ST}$ statistics and $F_4$ statistics of the form $F_4$(Yoruba, Test; WBI, CNE) (Extended Data Fig. 1 and 2). The resulting ancestry estimates for

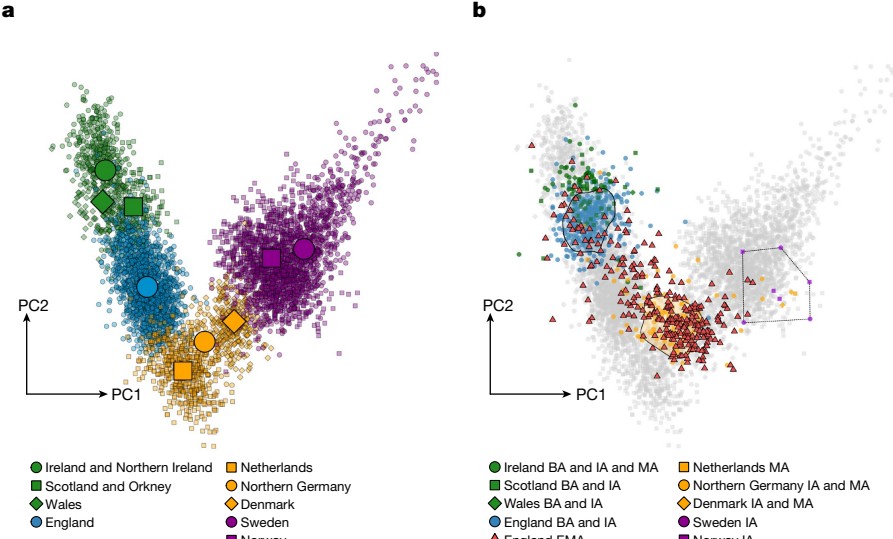

**Fig. 2 | PCA. a,** Present-day genomes from northwestern Europe. **b,** Published and novel ancient individuals in this study, projected onto **a.** Polygons indicate where two-thirds of the respective groups are located (England Bronze Age (BA) + Iron Age (IA) and North Sea IA + Early Middle Ages (EMA), respectively). The Scandinavian IA samples are connected with lines for clarity. For rough time boundaries of the samples used here, see Methods.

early medieval English individuals are indeed tightly congruent with both PCA PC1 position and $F_4$ statistics (Pearson's $|r| > 0.9$ between PCA, $F_4$ and ADMIXTURE ancestry assessments).

Applying our CNE–WBI ancestry decomposition to prehistoric samples, we found the genome-wide CNE ancestry in Britain and Ireland to be very low before the Early Middle Ages (Extended Data Fig. 3). In Bell Beaker and Bronze Age individuals from England, CNE ancestry does not account for more than 1% (Fig. 3a). This cannot be explained by genetic drift due to the temporal gap between our present-day CNE proxy and the Bronze Age, as shown by $F_4$ statistics (Extended Data Figs. 1 and 2), which are robust against such drift. Similar proportions were also measured during the Iron Age (1% on average). CNE ancestry increased only during the Roman period, to 15% on average, although this estimate is based on only seven individuals. Six of those seven Roman-era individuals are from a single site, *Eboracum* (present-day York); which was a *Colonia*, the highest rank of Roman city with a legionary fortress, and as such it may have attracted a more cosmopolitan population than most of the rest of England (Fig. 3b).

In contrast to these previous periods, the majority of the early medieval individuals from England in our sample derive either all or a large fraction of their ancestry from continental northern Europe, with CNE ancestry of 76 ± 2% on average (Methods). Although CNE ancestry is predominant in central and eastern England, it is much less prevalent in the south and southwest of England, and absent in the one site that we analysed from Ireland (Fig. 3b). Moreover, we observed differences in continental ancestry not only between but also within sites. Although we estimate CNE ancestry to be prevalent across eastern English cemeteries, there was considerable variation at the individual level, ranging from 0% to 100% of CNE ancestry within a site. For example, at Hatherdene Close ($n = 17$) in Cambridgeshire, we estimated a mean CNE ancestry of approximately 70%, with eight individuals exhibiting exclusively CNE ancestry, but three individuals having low or zero CNE ancestry. Overall, these patterns of genetic heterogeneity, from the transregional to the family level, are consistent with continuous interaction between the Iron Age-derived Romano-British population and migrants from the continent.

We find no significant differences of CNE or WBI ancestry between male and female individuals (Supplementary Note 7), and find individuals of both ancestries within prominent and/or furnished burials.

In England overall, individuals with CNE ancestry (here and in the following, CNE means more than 50% CNE, and WBI means less than 50% CNE) are more likely to be found with grave goods than individuals with WBI ancestry (Fisher's exact test $P = 0.016$). Of note, this appears to be driven by female individuals with CNE ancestry who are more likely to be found with grave goods ($P = 0.001$), and in particular brooches ($P = 0.012$), than female individuals with WBI ancestry (both based on Fisher's exact test). However, graves belonging to male individuals with CNE ancestry are just as likely to have grave goods ($P = 0.57$) or weapons ($P = 1$) as those with WBI ancestry (both based on Fisher's exact test). This is underlined by specific examples, such as a near 100% WBI male burial in grave 37 at Updown Eastry found with a seax under a barrow marked by a ring ditch, indicating a prominent weapon burial associated with a prominent person or status (Supplementary Fig. 1.1).

This pattern is also visible in East Anglia specifically, where individuals with CNE ancestry more often have grave goods ($P = 0.014$). This is also significant when considering only female individuals ($P = 0.025$), but not when considering females with brooches, which display gender-related status ($P = 0.197$). At the site level, these patterns are partly significant at Hatherdene Close ($P = 0.015$, $0.036$ and $0.1$, respectively). Treating ancestry not as a binary but as a continuous variable largely agrees with the previous results (see Supplementary Note 7), with a notable exception of West Heslerton, which stands out from this overall pattern, where men with a greater proportion of CNE ancestry are more likely to be found with weapons (Wilcoxon rank sum $P = 0.02$, although non-significant with Fisher's exact test $P = 0.53$), which is the only significant signal of this type that we found (Lakenheath also displays many CNE burials with weapons, but with limited sample size).

There are notable individual exceptions to these patterns, such as a predominantly (60%) WBI burial at grave 80 in Oakington, found with the skeleton of a cow, silvered disc brooches and a chatelaine, and interred under a barrow, which is one of the more notable or wealthy burials in this cemetery[49] (Supplementary Fig. 1.4). We note that several burials with weapons that were previously identified as female and discussed in the literature[23,50] have turned out to be genetically male in our analysis (see the highlighted entries in Supplementary Table 1). Of note, however, a single individual still displays a sex–gender difference: a teenage boy buried with an equal-arm brooch, beads and a knife (grave 122 in West Heslerton).

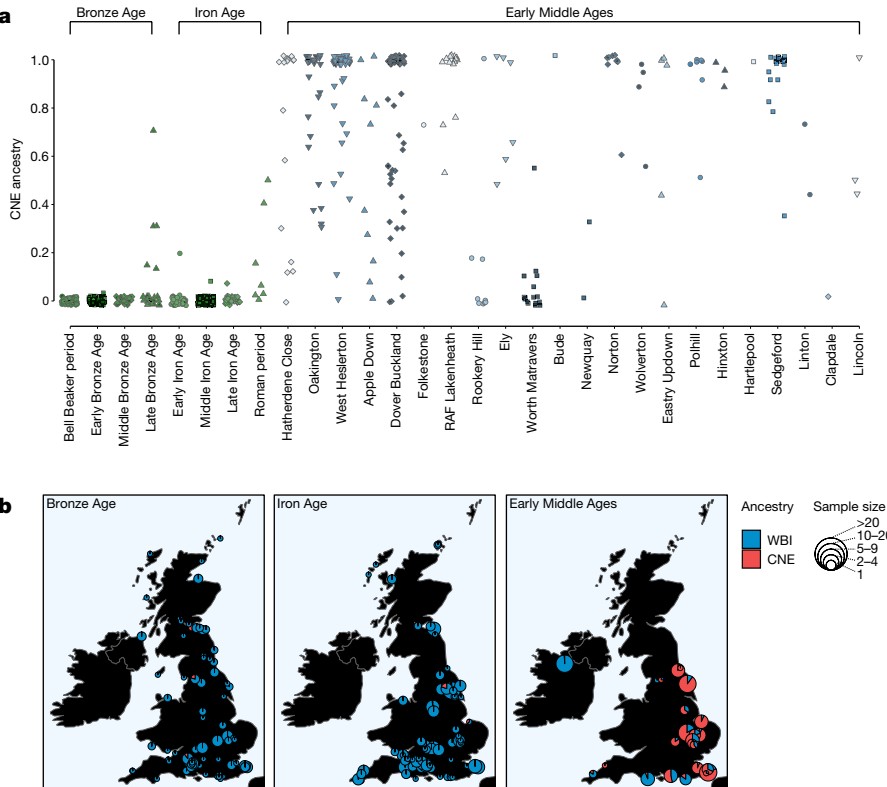

**Fig. 3 | Individual-based and site-based ancestry decomposition.**
**a**, Individual supervised admixture results for Bronze Age (*n* = 140), Iron Age (*n* = 304) and Early Middle Age (*n* = 285) genomes from England. The symbols and colours of early medieval sites correspond to Fig. 1. **b**, Mean CNE and WBI ancestry estimates of British–Irish sites from the Bronze, Iron and Early Middle Ages.

In Dover Buckland, one of the most comprehensively sampled cemeteries in our dataset, we observed the mixing of genetic and cultural identities at the family level. For example, we found a group of relatives, spanning at least three generations, who all exhibit unadmixed CNE ancestry (Extended Data Fig. 4a,c). Down the pedigree, we then see the integration of a female into this group, who herself had unadmixed WBI ancestry (grave 304), and two daughters (graves 290 and 426), consequently of mixed ancestry. WBI ancestry entered again one generation later, as visible in near 50:50 mixed-ancestry grandchildren (graves 414, 305 and 425). Grave goods, including brooches and weapons, are in fact found on both sides of this family tree, pre-mixing and post-mixing (for example, in the youngest and mixed generation, we found both weapons, beads and pin, and their mother with a brooch). Although the first mixed generation is buried in close proximity to each other, the grandchildren are elsewhere on the site, although placed together (Extended Data Fig. 4b).

A quite different pattern is observed at Apple Down, which is among the most western sites that we have analysed. Here graves can be classified into distinct burial configurations according to orientation, location and frequency of artefacts. We found that burials with CNE ancestry are more often buried in configuration A (located towards the middle of the site and with east–west burial orientation) than in configuration B (located more towards the edges and with north–south orientation)[49] (Fisher's exact test *P* = 0.048). This shows that there is a significant difference within the treatment of individuals according to their ancestry, a finding very similar to those at early medieval cemeteries in Hungary and Italy with respect to northern versus southern European ancestry[51].

## Ancestry sources across the North Sea

Our new continental medieval data from regions bordering the North Sea provide a unique opportunity to further investigate the potential source of the CNE-related ancestry increase that we have described above (Supplementary Note 3). To this end, we first selected individuals who, according to our CNE–WBI decomposition, are of unadmixed CNE ancestry (CNE of more than 95%; from here from as England EMA CNE). For each site in the continental dataset, we then tested whether its individuals were genetically similar to the England EMA CNE group (*n* = 109) in terms of allele frequencies. Among the continental medieval groups analysed, sites from both northern Germany and Denmark are indeed indistinguishable from England EMA CNE individuals (Fig. 4). Consistently, England EMA CNE and medieval individuals from Lower Saxony exhibit almost identical genetic affinities and ancestry components (Extended Data Fig. 5 and Supplementary Fig. 3.2), possess the highest level of genetic similarity (based on $F_2$, $F_3$, $F_4$ and $F_{ST}$ statistics) (Extended Data Fig. 5 and Supplementary Fig. 3.8) and are symmetrically related to most ancient and modern populations (Supplementary Table 3.12). Together, this suggests that they are likely derived from the same source population. Using the LOCATOR[52] software, which uses machine learning to map individuals into geographical space based on their genetic profiles, we infer a region spanning the northern Netherlands to the southernmost tip of Sweden as a putative source for the England EMA CNE ancestors, with a large proportion of individuals being assigned to Lower Saxony (see Methods) (Fig. 4). This similarity adds to previous evidence from the material culture and burial practices, especially between the Elbe-Weser region and the early Anglo-Saxon cemeteries, from which the archaeological migration discourse initially arose[53]. However, we also note the strong genetic homogeneity among most analysed sites in the northern Netherlands, northern Germany and Denmark (Supplementary Note 4), implying that, during the Early Middle Ages, the continental North Sea and adjacent western Baltic Sea area was a genetic continuum spanning most of the western North European plain without major geographical

substructure (Supplementary Fig. 4.1,4.4). This, together with genetic backflow from the British–Irish Isles into continental Europe (Supplementary Table 4.2 and Supplementary Fig. 4.2,4.4), reflects the inferred linguistic history[54] and precludes further identification of specific microregions that contributed gene flow to Britain. We note that, although our screening of plausible medieval continental sites is broad, it could overemphasize later developments of the genetic structure due to the increased replacement of cremation burials by inhumations on the continent. It also has a specific caveat in Scandinavia, where our medieval reference populations are mostly from Viking-era burials, which have diverse and mixed ancestries that may not be representative of the earlier populations there[42,44].

Already during the Early Middle Ages, several individuals from multiple sites exhibit modest degrees of excess affinity (5.4%) to present-day individuals from the Scandinavian peninsula (Supplementary Fig. 6.2a), indicating additional sources. Although close cultural contacts to the Scandinavian peninsula are attested in the archaeological record[3], we did not find this genetic variation to be geographically stratified within early medieval England (Supplementary Fig. 6.2b). This Scandinavian Peninsula-related ancestry increases substantially (to 30.6%) only during the Viking period (Supplementary Note 6).

Having established the close relatedness between specific continental regions and the individuals from early medieval England, we modelled the latter more directly using ancient source populations with the method qpAdm[35]. Specifically, we pooled ancient individuals in England by site and modelled each group as being admixed between two sources: one represented by pooled Iron Age/Roman period individuals from England, and the other represented by pooled early medieval individuals from Lower Saxony (from here known as LowerSaxony EMA). The resulting admixture proportions obtained in this way for early medieval sites in England are strongly correlated with our mean estimates from supervised ADMIXTURE above, as well as site-wise $F_4$ statistics and mean PCA position (Pearson's $|r| > 0.9$ between all four ancestry assessments) (Extended Data Fig. 6).

Using this model, we detected an average of 86 ± 2% ancestry from Lower Saxony across all early medieval sites in England, only slightly higher than the 76 ± 2% estimated using present-day source populations and supervised ADMIXTURE. At a regional scale, we observed more ancestry from Lower Saxony in eastern England than in the southwest, consistent with ancestry arriving from the east, either in one event or over a continuous time period. Our estimate of genome-wide ancestry is supported by independent evidence for population turnover from uniparental markers (Supplementary Fig. 2.7). Before the Middle Ages, post-Neolithic individuals from Britain and Ireland carried overwhelmingly the major Y chromosomal haplogroup R1b-P312, especially the sub-haplogroup R-L21 (refs. [39,41]), which in present day shows a cline across the region, with highest frequencies in the west[55,56]. By contrast, the early medieval population of England exhibits a substantial fraction of continental-derived haplotypes belonging to haplogroups R1b-U106, R1a-M420, I2a1-L460 and I1-M253, which are commonly found in northern and central Europe (and are also common among ancient continental individuals including the ones that we report). In particular, Y chromosomal haplogroups I1-M253 and R1a-M420 were absent from our Bronze, Iron and Roman Age British and Irish individuals, but were identified in more than one-third of our individuals from early medieval England. Overall, haplogroups absent in Bronze and Iron Age England represent at least 73 ± 4% of the Y chromosomes in our early medieval English sample, mirroring the turnover estimates from autosomal data. Similarly, mitochondrial genomes show evidence of female lineage population turnover from regions bordering the North Sea (Supplementary Note 2 and Supplementary Fig. 2.4).

Estimates of continental ancestry on the X chromosomes (Supplementary Fig. 2.11), as well as estimates of source origin of Y chromosomal haplogroups (Supplementary Fig. 2.16) point to no significant difference between male-specific lineages and autosomal admixture

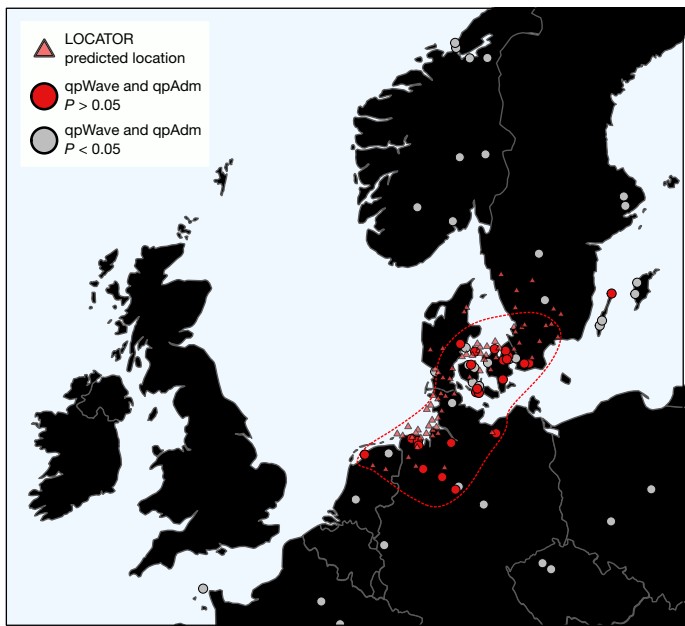

**Fig. 4 | Identifying continental source regions for immigrant ancestry in early Medieval England.** Shown are (1) continental sites that are genetically indistinguishable from the more than 95% CNE EMA English (England EMA CNE) population using qpWave and provide fitting *P* values as source in a two-way qpAdm model of England EMA, as well as (2) the predicted locations for 72 England EMA CNE genomes using LOCATOR[52]. The red dashed line marks where 95% of the qpAdm and qpWave data are located.

estimates (Supplementary Note 2). Although neither mitochondrial, Y chromosomal or X chromosomal data can exclude subtle levels of sex bias during the admixture (Supplementary Note 2), they are also consistent with a model of no sex bias, suggesting that the migrants included both men and women who mixed at similar levels with the local population. We note that absence of sex bias during the early medieval CNE–WBI admixture does not exclude possibilities for sex bias in the later admixture processes that caused the dilution of CNE ancestry in present-day England observed below.

## Recent population shifts in England

Although the most prominent signal of admixture in early medieval England is the rise in ancestry related to medieval and modern continental northern Europe, we found that several English sites include genomes that could not be explained as products of admixture between the two hypothesized ancestral gene pools—England IA or LowerSaxony EMA—using qpAdm[57]. Instead, these genomes have additional continental western and southern European ancestry (Supplementary Note 5). This ancestry is genetically very similar to Iron Age genomes from France[5,6] (France IA) (Extended Data Fig. 7, Supplementary Table 5.1 and Supplementary Fig. 5.3,5.4). The majority of this French Iron Age-derived ancestry is found in early medieval southeastern England, namely, at the sites of Apple Down, Eastry, Dover Buckland and Rookery Hill, where it constitutes up to 51% of the ancestry identified (Extended Data Fig. 8a and Supplementary Table 5.4).

The appearance of France IA-related ancestry in early medieval England anticipates a pattern that we also clearly see in the present-day English population structure, in which we found that the same two-way CNE–WBI model that fits most ancient English fails for the modern population (Supplementary Fig. 5.8,5.11). Indeed, the missing component in the modern English population appears to be represented well by France IA (Supplementary Table 5.2 and Supplementary Fig. 5.2).

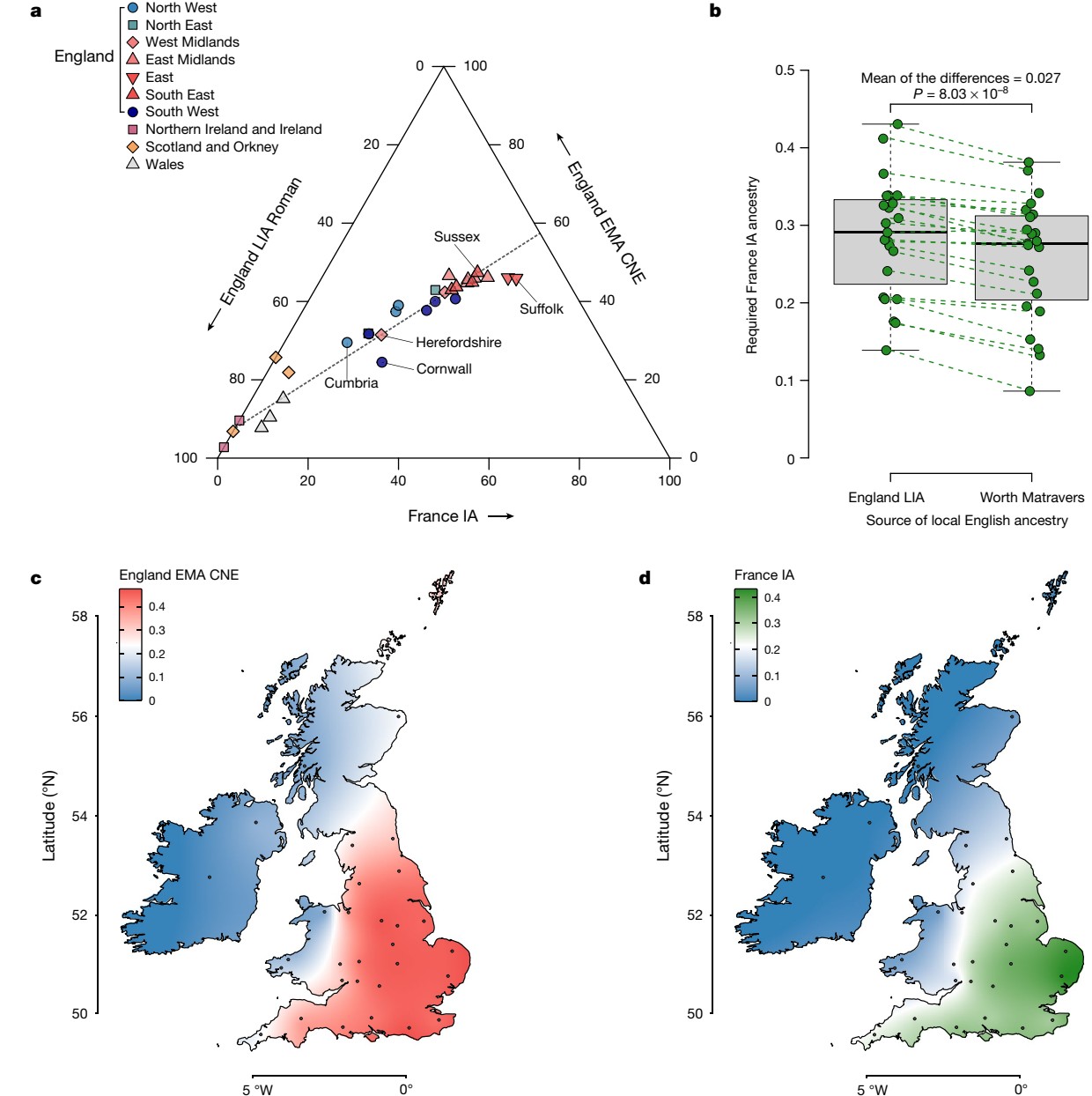

**Fig. 5 | Population structure of present-day Britain and Ireland. a**, Ternary plot of present-day British–Irish populations as a three-way admixture between late Iron Age and Roman England (England LIA Roman) (*n* = 32), France IA (*n* = 26) and England EMA CNE (*n* = 109). **b**, Boxplot comparison of France IA ancestry proportions in 23 English PoBI sampling regions using either England LIA Roman (*n* = 32) or Worth Matravers (*n* = 16) as source for local British ancestry in qpAdm. The *P* value obtained from a two-sided paired

Student's *t*-test is shown. The bounds of the box represent the 25th and 75th percentile, the centre represents the median, and the whiskers represent the minimum and maximum values in the data. Dashed lines connect points from the same region. **c**, Geographical distribution of the England EMA CNE, ancestries based on the interpolation of 31 present-day population estimates. The coordinates of the sample collection districts approximate the centroids of the averaged birthplaces of the grandparents. **d**, Same as **c**, but for France IA.

Using qpAdm (Methods), most present-day Scottish, Welsh and Irish genomes can be modelled as receiving most or all of their ancestry from the British Bronze or Iron Age reference groups, with little or no continental contribution. By contrast, for all present-day English samples the simple two-way admixture model (England LIA + England EMA CNE) fails. By extending our model to a three-way with added France IA as a third component, we now obtain fitting models (Supplementary Fig. 5.11,5.21). We estimate that the ancestry of the present-day English ranges between 25% and 47% England EMA CNE-like, 11% and 57% England LIA-like and 14% and 43% France IA-like. There are substantial genetic differences between English regions (Fig. 5a), with less ancient continental ancestry (England EMA CNE or France IA related) evident

in southwestern and northwestern England as well as along the Welsh borders (Fig. 5c). By contrast, we saw peaks in CNE-like ancestry of up to 47% for southeastern, eastern and central England, especially Sussex, the East Midlands and East Anglia. We found substantial France IA ancestry only in England, but not in Wales, Scotland or Ireland, following an east-to-west cline in Britain (Pearson's |*r*| > 0.86), accounting for as much as 43% of the ancestry in East Anglia (Fig. 5d). Very similar results were produced using LowerSaxony EMA as a source for CNE ancestry (Extended Data Fig. 8b). One potential caveat in this analysis is our relatively sparse Roman sample from England, where we particularly lack samples from the south, which might have pre-existing France IA-related ancestry. We, therefore, turned to one of our early medieval

sites, the post-Roman cemetery of Worth Matravers at the southern coast of Dorset, whose individuals have nearly no CNE ancestry (less than 6% on average), and thus may serve as a more temporally close proxy for post-Roman Britain before the arrival of CNEs. When used as a source in our model, we found that the estimates of France IA-related ancestry in present-day England changed by less than 3% on average across the regions (Fig. 5b), suggesting that France IA-related ancestry entered England to a substantial amount after the Roman period. We note that a model involving southern or western European-like ancestry in England has been previously proposed[36] on the basis of present-day samples, but we can now go further and delineate this third component more clearly against the CNE-like immigrant gene pool making up the majority of the early medieval individuals from England that we studied.

Our three-way population model for present-day England supports a view of post-Roman English genetic history as punctuated by gene flow processes from at least two major sources: first, the attested arrival of CNE ancestry during the Early Middle Ages from northern Germany, the Netherlands and Denmark, and second, the arrival of ancestry related to France IA. Although we cannot precisely date the order of those arrivals, at least substantial amounts of France IA-related ancestry seem to be absent in northern and eastern England during the Early Middle Ages and therefore must have arrived there subsequently. In other parts of England, however, it may have entered together with CNE ancestry or even earlier. Notably in southern England, namely, Eastry, Apple Down and Rookery Hill, several early medieval individuals already exhibit France IA-related ancestry, which probably results, at least in part, from localized mobility between the south of England and the Frankish areas of Europe during the Early Middle Ages (Extended Data Fig. 8a). Indeed, Frankish material culture is evident in these regions, particularly in Kent and Sussex[58–60]. Admixture from this second source is, therefore, unlikely to have resulted from a single discrete wave. More plausibly, it resulted from pulses of immigration or continuous gene flow between eastern England and its neighbouring regions.

## Discussion

The 'Anglo-Saxon settlement' is among the most intensely debated topics in British history, but much of the discussion remains anchored to the contents of Bede's *Ecclesiastical History* and the *Anglo-Saxon Chronicle*[18]. These early writings defined the settlement as a single event, or a series of events, tied to the immediate aftermath of the Roman administration in the fifth to sixth century. In the archaeological and historical debate, this has been described as happening to varying degrees; as the *Adventus Saxonum* (a folk migration of named Germanic tribes), an invasion or the movement of a limited number of elite male migrants[18,61]. To this day, little agreement has been reached over the scale of migration, the mode of interaction between locals and newcomers, or how the transformation of the social, material, and linguistic or religious spheres was achieved. Here we provide strong evidence of large-scale early medieval migration across the North Sea zone and extend its temporal scope. In particular, we show that these migrations started earlier than previously assumed, as evidenced by individuals with CNE ancestry from later Roman contexts, and continued throughout the middle Anglo-Saxon period. Our results from middle Saxon sites such as Sedgeford push the estimated dates of arrival of CNE ancestry to as late as the eight century and merge these events with interpersonal mobility from Sweden and other Scandinavian regions during the later Viking invasion and settlement. Together, these migrations appear to be part of a continuous movement of people from across the North Sea to Britain from the later Roman period into the eleventh century CE.

Our results overwhelmingly support the view that the formation of early medieval society in England was not simply the result of a small elite migration[18,61], but that mass migration from afar must also have had a substantial role. We identified numerous individuals with only continental ancestry, suggesting that many of them were migrants themselves or were their unadmixed descendants. Both the lack of genetic evidence for male sex bias, and the correlation between ancestry and archaeological features, point to women being an important factor in this migration. Although men with migrant and local ancestry were buried in similar ways, women with migrant ancestries were more often found with grave goods than women with local ancestry. This could point to social stratification, or plausibly might simply reflect the degree to which women of local ancestry were integrated into the emerging CNE families. It is clear, however, that these social differences are subtle, given that we did not find evidence for this pattern in male burials, and that we found significant regional and site-level differences. Previous hypotheses about the social mechanisms in this migration have included partial social segregation[62], elite migration[18,61], substantial population replacement[34] or no migration at all[1,22]. Our combined genetic and archaeological analysis point to a complex, regionally contingent migration with partial integration that was probably dependent on the fortunes of specific families and their individual members.

In present-day Britain, we saw substantial northern continental ancestry, albeit at a lower level than during the early medieval period, pointing to a lasting demographic impact of the 'Anglo-Saxon' migrations. Specifically, in early medieval western England, Wales and Scotland, and more generally in England during the Norman period, further aDNA sampling may clarify how CNE ancestry spread and was subsequently diluted. Beyond the substantial early medieval immigration of northwestern continental European people found here, we have also identified a second major source of continental ancestry in modern Britain from sources more to the European south and west. This second ancestry component is already evident in our early medieval samples. In Southeast England specifically, individuals at several sites show ancestry whose closest match is in modern-day western Germany, Belgium and/or France, which matches the Frankish connections seen in the archaeological record for these regions. Our data and analyses indicate that this second genetic introgression continued further into the Middle Ages and potentially beyond.

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

Joscha Gretzinger[1], Duncan Sayer[2✉], Pierre Justeau[3], Eveline Altena[4], Maria Pala[3], Katharina Dulias[3,54], Ceiridwen J. Edwards[3,5], Susanne Jodoin[6], Laura Lacher[1], Susanna Sabin[7], Åshild J. Vågene[8], Wolfgang Haak[1], S. Sunna Ebenesersdóttir[9,10], Kristjan H. S. Moore[9], Rita Radzeviciute[1], Kara Schmidt[11], Selina Brace[12], Martina Abenhus Bager[8,14], Nick Patterson[13,14], Luka Papac[1], Nasreen Broomandkhoshbacht[13,15], Kimberly Callan[13,15], Éadaoin Harney[13], Lora Iliev[13,15], Ann Marie Lawson[13,15], Megan Michel[1,13,15], Kristin Stewardson[13,15], Fatma Zalzala[13,15], Nadin Rohland[13,14], Stefanie Kappelhoff-Beckmann[16], Frank Both[16], Daniel Winger[17], Daniel Neumann[18], Lars Saalow[19], Stefan Krabath[20], Sophie Beckett[21,22,23], Melanie Van Twest[21], Neil Faulkner[21], Chris Read[24], Tabatha Barton[25], Joanna Caruth[26], John Hines[27], Ben Krause-Kyora[28], Ursula Warnke[16], Verena J. Schuenemann[29,30,31], Ian Barnes[12], Hanna Dahlström[32], Jane Jark Clausen[32], Andrew Richardson[33,34], Elizabeth Popescu[35], Natasha Dodwell[35], Stuart Ladd[35], Tom Phillips[35], Richard Mortimer[35,55], Faye Sayer[36], Diana Swales[37], Allison Stewart[2], Dominic Powlesland[38], Robert Kenyon[39], Lilian Ladle[40], Christina Peek[20], Silke Grefen-Peters[41], Paola Ponce[42], Robin Daniels[43], Cecily Spall[44], Jennifer Woolcock[45], Andy M. Jones[46], Amy V. Roberts[47], Robert Symmons[48], Anooshka C. Rawden[48,49], Alan Cooper[50], Kirsten I. Bos[1], Tom Booth[51], Hannes Schroeder[8], Mark G. Thomas[52], Agnar Helgason[9,10], Martin B. Richards[3], David Reich[13,14,15,53], Johannes Krause[1] & Stephan Schiffels[1✉]

[1]Max Planck Institute for Evolutionary Anthropology, Leipzig, Germany. [2]University of Central Lancashire, Preston, UK. [3]University of Huddersfield, Huddersfield, UK. [4]Leiden University, Leiden, Netherlands. [5]University of Oxford, Oxford, UK. [6]University of Tübingen, Tübingen, Germany. [7]Center for Evolution and Medicine, Arizona State University, Tempe, AZ, USA. [8]Globe Institute, Faculty of Health and Medical Sciences, University of Copenhagen, Copenhagen, Denmark. [9]deCODE Genetics/AMGEN Inc., Reykjavík, Iceland. [10]Department of Anthropology, School of Social Sciences, University of Iceland, Reykjavík, Iceland. [11]University of Münster, Münster, Germany. [12]Department of Earth Sciences, Natural History Museum, London, UK. [13]Department of Genetics, Harvard Medical School, Boston, MA, USA. [14]Broad Institute of Harvard and MIT, Cambridge, MA, USA. [15]Howard Hughes Medical Institute, Harvard Medical School, Boston, MA, USA. [16]Landesmuseum Natur und Mensch, Oldenburg, Germany. [17]University of Rostock, Rostock, Germany. [18]Lower Saxony State Museum, Hanover, Germany. [19]Landesamt für Kultur und Denkmalpflege Mecklenburg-Vorpommern, Schwerin, Germany. [20]Institute for Historical Coastal Research (NIhK), Wilhelmshaven, Germany. [21]Sedgeford Historical and Archaeological Research Project, Sedgeford, UK. [22]Cranfield Forensic Institute, Cranfield Defence and Security, Cranfield University, Cranfield, UK. [23]Melbourne Dental School, University of Melbourne, Melbourne, Victoria, Australia. [24]The Atlantic Technological University, Sligo, Ireland. [25]Milton Keynes Museum, Milton Keyes, UK. [26], Cotswold Archaeology, Needham Market, UK. [27]Cardiff University, Cardiff, UK. [28]University of Kiel, Kiel, Germany. [29]University of Zurich, Zurich, Switzerland. [30]Department of Evolutionary Anthropology, University of Vienna, Vienna, Austria. [31]Human Evolution and Archaeological Sciences, University of Vienna, Vienna, Austria. [32]Museum of Copenhagen, Copenhagen, Denmark. [33]Canterbury Archaeological Trust, Canterbury, UK. [34]Isle Heritage CIC, Sandgate, UK. [35]Oxford Archaeology East, Cambridge, UK. [36]University of Birmingham, Birmingham, UK. [37]Centre for Anatomy and Human Identification (CAHID), University of Dundee, Dundee, UK. [38]The Landscape Research Centre Ltd, Yedingham, UK. [39]East Dorset Antiquarian Society (EDAS), West Bexington, UK. [40]Department of Archaeology and Anthropology, Bournemouth University, Poole, UK. [41]Ossatura–Wilhelm-Börker, Braunschweig, Germany. [42]University of York, York, UK. [43]Tees Archaeology, Hartlepool, UK. [44]FAS Heritage, York, UK. [45]Royal Cornwall Museum, Truro, UK. [46]Cornwall Archaeological Unit, Truro, UK. [47]The Novium Museum, Chichester, UK. [48]Fishbourne Roman Palace, Fishbourne, UK. [49]South Downs Centre, Midhurst, UK. [50]BlueSkyGenetics, Adelaide, South Australia, Australia. [51]Natural History Museum, London, UK. [52]University College London, London, UK. [53]Department of Human Evolutionary Biology, Harvard University, Cambridge, MA, USA. [54]Present address: Institute of Geosystems and Bioindication, Technische Universität Braunschweig, Braunschweig, Germany. [55]Present address: Cotswold Archaeology, Needham Market, UK. [56]Deceased: Neil Faulkner. ✉e-mail: dsayer@uclan.ac.uk; stephan_schiffels@eva.mpg.de

## Methods

### Study design

**Archaeological research.** Provenance information for samples from all archaeological sites are given in Supplementary Information Section 1, together with short descriptions of each site, the institution owning the samples (or custodians of the samples), the responsible coauthor who obtained permission to analyse and the year of the permission granted.

**Sampling.** Sampling of 494 bone and teeth samples took place in clean-room facilities dedicated to aDNA work, for 296 samples at the Max Planck Institute for Science of Human History in Jena (MPI-SHH), for 65 at the Department of Biological and Geographical Sciences at the University of Huddersfield, for 33 at the Department of Genetics, Harvard Medical School (HMS), for 32 at the Institute for Scientific Archaeology of the Eberhard Karls University Tübingen, for 31 at the Leiden University Medical Centre in Leiden, for 15 at the Globe Institute of the University of Copenhagen, for 12 at the Australian Centre for Ancient DNA at the University of Adelaide, and for 10 at the Research Laboratory for Archaeology, University of Oxford. The sampling workflow included documenting and photographing the provided samples. For teeth processed at the MPI-SHH, we cut along the cementum–enamel junction and collected powder by drilling into the pulp chamber. The teeth processed at the Leiden University Medical Centre were sampled according to a previously published paper[63]. For the petrous bones, we either cut the petrous pyramid longitudinally to drill the dense part directly from either side[64] or applied the cranial base drilling method as previously described[65]. We collected between 30 and 200 mg of bone or tooth powder per sample for DNA extractions.

**DNA extraction.** At MPI-SHH and HMS, aDNA was extracted following a modified protocol[66], as described in www.protocols.io/view/ancient-dna-extraction-from-skeletal-material-baksicwe, in which we replaced the extended-MinElute-column assembly for manual extractions with columns from the Roche High Pure Viral Nucleic Acid Large Volume Kit[67], and for automated extraction with a protocol that replaced spin columns with silica beads in the purification step[68]. Extraction of aDNA at the Leiden University Medical Centre, at the Globe Institute and at the University of Adelaide followed the protocols of Kootker et al.[63], Damgaard et al.[69] and Brotherton et al.[70], respectively. Extraction of aDNA at the Universities of Oxford and Huddersfield followed a published protocol[71].

**Library construction.** We generated 104 double-indexed[72] double-stranded libraries using 25 µl of DNA extract and following established protocols[73]. We applied the partial UDG (UDG half)[74] protocol to remove most of the aDNA damage while preserving the characteristic damage pattern in the terminal nucleotides. For 375 extracts, we generated double-indexed single-stranded libraries[75] using 20 µl of DNA extract and applied no UDG treatment.

**Shotgun screening, capture and sequencing.** Libraries produced at MPI-SHH were sequenced in-house on an Illumina HiSeq 4000 platform to an average depth of 5 million reads and after demultiplexing processed through EAGER[76]. After an initial quality filter based on the presence of aDNA damage and endogenous DNA higher than 0.1%, we subsequently enriched 439 libraries using in-solution capture probes synthesized by Agilent Technologies for approximately 1240K SNPs along the nuclear genome[77]. The captured libraries were sequenced for 20–40 million reads on average using either a single end (1 × 75 bp reads) or paired-end configuration (2 × 50 bp reads). In addition, 40 genomes were shotgun sequenced for 225 million reads on average to low coverage. For the 120 samples processed at HMS, we enriched for sequences overlapping approximately 1240K SNPs[77] as well as the mitochondrial genome[78], and sequenced on Illumina NextSeq500 instruments for 2 × 76 cycles, or on HiseqX10 instruments for 2 × 101 cycles (reading out both indices) approximately until the point in which every additional 100 sequences generated yielded fewer than one additional SNP with data.

### aDNA data processing

**Read processing and aDNA damage.** For data produced at the MPI-SHH, after demultiplexing based on a unique pair of indexes, raw sequence data were processed using EAGER[76]. This included clipping sequencing adaptors from reads with AdapterRemoval (v2.3.1)[79] and mapping of reads with Burrows–Wheeler Aligner (BWA)[80] v0.7.12 against the human reference genome hg19, with seed length (-l) disabled, maximum number of differences (-n) of 0.01 and a quality filter (-q) of 30. We removed duplicate reads with the same orientation and start and end positions using DeDup[76] v0.12.2. Terminal base deamination damage calculation was done using mapDamage[81] v2.0.6, specifying a length (-l) of 100 bp. For the 107 libraries that underwent UDG half treatment, we used BamUtil v1.0.14 (https://genome.sph.umich.edu/wiki/BamUtil:_trimBam) to clip two bases at the start and end of all reads for each sample to remove residual deaminations, thus removing genotyping errors that could arise due to aDNA damage.

For data produced at the HMS, after trimming barcodes and adapters[57], we merged read pairs with at least 15 bp of overlap, allowing no more than one mismatch if base quality was at least 20, or up to three mismatches if base qualities were less than 20. We chose the nucleotide of the higher quality in case of a conflict while setting the local base quality to the minimum of the two using a custom toolkit (https://github.com/DReichLab/ADNA-Tools). We aligned merged sequences to human genome hg19 using BWA[80] v0.7.15 with a maximum number of differences (-n) of 0.01, a maximum number of gap opens (-o) of 2 and seed length (-l) of 16,500. PCR duplicates were identified by tagging all aligned sequences with the same start and stop positions and orientation and, in some cases, in-line barcodes using Picard MarkDuplicates (http://broadinstitute.Github.io/picard/). We only considered sequences that spanned at least 30 bp, and subsequently selected a single copy of each such sequence that had the highest base-quality score. To remove aDNA damage, we trimmed the last two bases of each sequence for UDG-treated libraries and the last five for non-UDG-treated libraries.

**Sex determination.** To determine the genetic sex of ancient individuals processed at the MPI-SHH, we calculated the coverage on the autosomes as well as on each sex chromosome and subsequently normalized the X reads and Y reads by the autosomal coverage[82]. For that, we used a custom script (https://github.com/TCLamnidis/Sex.DetERRmine) for the calculation of each relative coverage as well as their associated error bars[83]. Female individuals were expected to have an X rate of 1 and a Y rate of 0, whereas male individuals were expected to have a rate of 0.5 for both X and Y chromosomes. For individuals processed at the HMS, we calculated the ratio of sequences mapping to the Y chromosome to the sum of sequences mapping to the X and Y chromosome for the 1240K data. A ratio less than 3% is consistent with a female individual and a ratio higher than 35% is consistent with a male individual[6].

**Contamination estimation.** We used the Analysis of Next Generation Sequencing Data (ANGSD) package[84] (v0.923) to test for heterozygosity of polymorphic sites on the X chromosome in male individuals, applying a contamination threshold of 5% at the results of method two. For male and female samples, we estimated contamination levels on the mitochondrial DNA either using Schmutzi[85] (v1.5.4) by comparing the consensus mitogenome of the ancient sample to a panel of 197 worldwide mitogenomes as a potential contamination source (MPI-SHH), or by estimating the match rate to the consensus sequence using contamMix[86] v1.0-12 (HMS), applying a contamination threshold of 5%. We

used PMDtools[87] (v0.50) to isolate sequences from each sample that had clear evidence of contamination (over 5% on the X chromosome or mitogenome) according to the post-mortem damage score (PMD score > 3, using only bases with phred-scaled quality of at least 30 to compute the score), and performed contamination estimation again. If a sample scored below the threshold, it was included in the analysis and modelling. If the authenticity of a sample could not be verified or falsified, it was included in population genetic analyses but not used for modelling. In summary, the median mitochondrial DNA contamination is 1.0%, and the median X chromosome contamination is 1.1% (after PMD filtering).

**Genotyping.** We used the program pileupCaller (v1.4.0.2) (https://github.com/stschiff/sequenceTools.git) to genotype the trimmed BAM files of UDG half libraries. A pileup file was generated using samtools mpileup with parameters -q 30 -Q 30 -B containing only sites overlapping with our capture panel. From this file, for each individual and each SNP on the 1240K panel[57,88,89], one read covering the SNP was drawn at random, and a pseudohaploid call was made, that is, the ancient individual was assumed homozygous for the allele on the randomly drawn read for the SNP in question. For libraries that underwent no UDG treatment, we used the parameter -SingleStrandMode, which causes pileupCaller to ignore reads aligning to the forward strand at C/T polymorphisms and at G/A polymorphisms to ignore reads aligning to the reverse strand, which should remove post-mortem damage in aDNA libraries prepared with the non-UDG single-stranded protocol.

**Mitochondrial and Y chromosome haplogroup assignment.** To process mitochondrial DNA data generated at the MPI-SHH, we extracted reads from 1240K data using samtools[90] v1.3.1 and mapped these to the revised Cambridge reference sequence. At the HMS, we aligned merged sequences to the mitochondrial genome RSRS[91]. We subsequently called consensus sequences using Geneious[92] R9.8.1 and used HaploGrep 2 (ref. [93]) v2.4.0 (https://haplogrep.uibk.ac.at/; with PhyloTree version 17-FU1) to determine mitochondrial haplotypes. For the male individuals processed at the MPI-SHH, we used pileup from the Rsamtools package to call the Y chromosome SNPs of the 1240K SNP panel (mapping quality of 30 or more and base quality of 30 or more). We then manually assigned Y chromosome haplogroups using pileups of Y SNPs included in the 1240K panel that overlap with SNPs included on the ISOGG SNP index v.15.73 (Y-DNA Haplogroup Tree 2019–2020; 2020.07.11). For male individuals processed at the HMS, we automatically determined Y chromosome haplogroups using both targeted SNPs and off-target sequences aligning to the Y chromosome based on comparisons to the Y chromosome phylogenetic tree from Yfull version 8.09 (https://www.yfull.com/)[6].

**Kinship estimation.** We calculated the PWMR[94] in all pairs of individuals from our pseudo-haploid dataset to double check for potential duplicate individuals and to determine first-degree, second-degree and third-degree relatives. For this purpose, we also used READ[95] to determine first-degree, second-degree and third-degree relatedness among individuals based on the proportion of non-matching alleles (P0) in nonoverlapping windows of 1 Mb and to calculate standard errors. We also used the method LcMLkin[96], which uses genotype likelihoods to estimate the three k-coefficients (k0, k1 or k2), which define the probability that two individuals have zero, one or two alleles identical by descendent at a random site in the genome. We performed LcMLkin to distinguish between possible parent–offspring or sibling relationships.

**Population genetic analysis**
**Dataset.** We merged our aDNA data with previously published datasets of 4,336 ancient individuals reported by the Reich laboratory in the Allen Ancient DNA Resource v.50.0 (https://reich.hms.harvard.edu/allen-ancient-dna-resource-aadr-downloadable-genotypes-present-day-and-ancient-dna-data). We assembled a dataset from mostly European populations for genome-wide analyses[36,45–47,97–103]. This modern set includes 10,176 individuals (Supplementary Note 2). Loci and individuals with less than 95% call rate as well as a 15-Mb region surrounding the HLA region[36] were removed and loci on three previously reported long-range linkage disequilibrium regions on chromosomes 6, 8 and 11 (refs. [104,105]) were pruned using PLINK[106] (v1.90b3.29). aDNA data were merged to this dataset, correcting for reference allele and strand flips. We kept 445,171 autosomal SNPs after intersecting autosomal SNPs in the 1240K capture with the modern analysis set.

**Abbreviations.** We used the following abbreviations in population labels: N, Neolithic; C, Chalcolithic; EBA, Early Bronze Age; MBA, Middle Bronze Age; LBA, Late Bronze Age; Iron Age, IA; RA, Roman Age; EMA, Early Middle Ages; MA, Middle Ages. In Britain, these periods roughly correspond to the following simplified time ranges: Neolithic: 4000–2500 BCE, Chalcolithic and EBA: 2500–1600 BCE; MBA: 1600–1200 BCE; LBA: 1200–800 BCE; IA: 800 BCE to 400 CE; EMA 400–1000 CE.

**PCA.** We carried out PCA using the smartpca software v16000 from the EIGENSOFT package (v6.0.1)[107]. We computed PCs on three different sets of modern European populations (Supplementary Note 2) and projected ancient individuals using lsqproject: YES.

**F statistics.** $F_3$ and $F_4$ statistics were computed with ADMIXTOOLS[108] v3.0 (https://github.com/DReichLab). $F_3$ statistics were calculated using qp3Pop (v435). For $F_4$ statistics, we used the qpDstat (v755) and with the activated $F_4$ mode. Significant deviation from zero can be interpreted as rejection of the tree population typology ((outgroup, X); (Pop1, Pop2)). Under the assumption that no gene flow occurred between Pop1 and Pop2 and the outgroup, a positive $f$-statistic suggests affinity between X and Pop2, whereas a negative value indicates affinity between X and Pop1. Standard errors were calculated with the default block jackknife 5 cM in size.

**Fixation index.** We calculated $F_{ST}$ using smartpca software v16000 from the EIGENSOFT package (v6.0.1)[107] with the fstonly, inbreed and fsthiprecision options set to YES.

**Maximum likelihood tree.** We constructed maximum likelihood trees using TreeMix (v1.12)[109]. For each tree, we performed a round of global rearrangements after adding all populations (-global) and calculated 100 bootstrap replicates to assess the uncertainty of the fitted model (-bootstrap). Sample size correction was disabled.

**Inference of mixture proportions.** We estimated ancestry proportions using qpWave[57,110] (v410) and qpAdm[57] (v810) from ADMIXTOOLS[108] v3.0 with the allsnps: YES option and a basic set of 11 outgroups: YRI. SG, Poland, Finland, Sweden, Denmark, Ireland, Wales, Italy, Spain, Belgium and the Netherlands. For some analyses, we added additional outgroups to this basic set (Supplementary Notes 4–6).

**Prediction of geographical origins.** LOCATOR[52] (v1.2) was run using a geolocated reference panel consisting of 670 Bronze Age, Iron Age and medieval European samples with 1X coverage higher than 50% and considering only polymorphisms covered at least in 50% of the samples, leaving a total of 920,060 SNPs. Default parameters were used, except that the width of each neural layer was 512 and -imputed-missing was set to YES. The best run was selected as the one showing the lowest validation error and the highest $R^2$ numbers from a total of 40 independent runs.

**ADMIXTURE analysis.** We performed model-based clustering analysis using ADMIXTURE[48] (v1.3). We used ADMIXTURE in supervised mode,

in which we estimated admixture proportions for the ancient individuals using modern reference populations at various $K$ values (Supplementary Notes 3–6). These analyses were run on haploid data with the parameter –haploid set to all (="*"). Standard errors for point estimates were calculated using 1,000 bootstrap replicates with the -B parameter. To obtain point estimates for populations, we averaged individual point estimates and calculated the standard error of the mean (s.e.m.): s.e.m. $= \frac{\sigma}{\sqrt{n}}$. We found that this better reflects the diversity within the population than a propagation of error approach, which underestimates the variance within the point estimate sample. For unsupervised ADMIXTURE analysis (Supplementary Notes 3 and 5), we carried out linkage disequilibrium pruning on the dataset using PLINK[106] with the flag –indep-pairwise 200 25 0.4, leaving 306,393 SNPs. We ran ADMIXTURE with the cross-validation (–cv.) flag specifying from $K = 2$ to $K = 10$ clusters, with five replicates for each value of $K$. For each value of $K$, the replicate with highest log likelihood was kept.

## Reporting summary

Further information on research design is available in the Nature Research Reporting Summary linked to this article.

## Data availability

Raw sequence data (bam files) from the 479 newly reported ancient individuals will be available before publication from the European Nucleotide Archive under accession number PRJEB54899. Published genotype data for the present-day British sample are available from the WTCCC via the European Genotype Archive (https://www.ebi.ac.uk/ega/) under accession number EGAD00010000634. Published genotype data for the present-day Irish sample are available from the WTCCC via the European Genotype Archive under accession number EGAD00010000124. Published genotype data for the rest of the present-day European samples are available from the WTCCC via the European Genotype Archive under accession number EGAD00000000120. Published genotype data for the Dutch samples are available by the GoNL request process from The Genome of the Netherlands Data Access Committee (DAC) (https://www.nlgenome.nl). The Genome Reference Consortium Human Build 37 (GRCh37) is available via the National Center for Biotechnology Information under accession number PRJNA31257. The revised Cambridge reference sequence is available via the National Center for Biotechnology Information under NCBI Reference Sequence NC_012920.1. Previous published genotype data for ancient individuals were reported by the Reich laboratory in the Allen Ancient DNA Resource v.50.0 (https://reich.hms.harvard.edu/allen-ancient-dna-resource-aadr-downloadable-genotypes-present-day-and-ancient-dna-data).

## Code availability

All software used in this work is publicly available. Corresponding publications are cited in the main text and Supplementary information. List of software and respective versions: AdapterRemoval (v2.3.1), Burrows–Wheeler Aligner (v0.7.12), DeDup (v0.12.2), mapDamage (v2.0.6), BamUtil (v1.0.14), EAGER (v1), Sex.DetERRmine (https://github.com/TCLamnidis/Sex.DetERRmine) (v1.1.2), ANGSD (v0.915), Schmutzi (v1.5.4), contamMix (v1.0-12), PMDtools (v0.50), pileupCaller (v1.4.0.2), samtools (v1.3.1), Geneious R9.8.1, HaploGrep 2 (v2.4.0), READ (https://bitbucket.org/tguenther/read) (vf541d55), lcMLkin (https://github.com/COMBINE-lab/maximum-likelihood-relatedness-estimation) (v0.5.0), PLINK (v1.90b3.29), Picard tools (v2.27.3), ADNA-Tools (https://github.com/DReichLab/ADNA-Tools) (v3b4357d), smartpca (v16000; EIGENSOFT v6.0.1), qp3Pop (v.435; ADMIXTOOLS v3.0), qpDstat (v.755; ADMIXTOOLS v3.0), Treemix (v1.12), qpWave (v410), qpAdm (v.810), LOCATOR (v1.2) and ADMIXTURE (v1.3). The code used in Supplementary Note 2 ('Estimating sex-biased ancestry from uniparental markers in the presence of variable admixture proportions') can be found at https://github.com/stschiff/AngloSaxon_Y-chrom_sex-bias. Data visualization and descriptive statistical tests were performed in R (v4.1.1). The following R packages were used: Rsamtools (v2.12.0), binom (v1.1-1.1), ape (v.5.6-2), phytools (v1.0-3), psych (v2.2.5), vegan (v2.6-2), factoextra (v1.0.7), ggplot2 (v3.3.6), ggExtra (v0.10.0), ggforce (v0.3.3), rnaturalearth (v0.1.0), sf (v1.0.–8), raster (v3.5-21), elevatr (v0.4.2), rgdal (v1.5-32), spatstat (v2.3-4), maptools (v1.1-4), gstat (v2.0-9), sp (v1.5-0), labdsv (v2.0-1), igraph (v1.3.4), magrittr (v2.0.3), dplyr (v1.0.9), reshape 2 (v1.4.4) and tidyverse (v.1.3.2). Y chromosomal and mitochondrial DNA haplogroups were determined using the ISOGG SNP index (v15.73) and PhyloTree (v17-FU1) reference databases, respectively.

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

**Acknowledgements** We thank the Brighton & Hove Museums and A. Maxted; the Stiftung Archäologie im rheinischen Braunkohlerevier; the Max Planck Society; N. Adamski and A. Claxton; and the Velux Foundations. This project has received funding from the European Research Council (ERC) under the European Union's Horizon 2020 research and innovation programme (grant agreement number 851511). P.J. and K.D. were supported by a Leverhulme Doctoral Scholarship awarded to M.B.R. and M.P. C.J.E. acknowledges support from The Leverhulme Trust via research project grant RPG-388. I.B., T. Booth, M.G.T. and S. Brace were supported by a Wellcome Trust Investigator Award (project number 100713/Z/12/Z). S. Beckett and M.V.T. thank R. Baldry and North West Norfolk History Society for funding of radiocarbon dating analyses. Sampling and DNA extraction of the samples from Groningen was funded by the European Funds for Regional Development and the Province of Groningen. We are grateful to the Museum für Archäologie in der Stiftung Schleswig-Holsteinische Landesmuseen Schloss Gottorf for providing samples from Schleswig Rathausmarkt. The ancient DNA laboratory work at Harvard University was supported by the US National Institutes of Health grant GM100233, by John Templeton Foundation grant 61220, by a gift from J.-F. Clin, by the Howard Hughes Medical Institute, and by the Allen Discovery Center program, a Paul G. Allen Frontiers Group advised program of the Paul G. Allen Family Foundation.

**Author contributions** E.A., C.J.E., S.J., L. Lacher, S. Sabin, K. Schmidt, M.A.B., S.K.-B., F.B., D.W., D.N., L.S., S.K., S. Beckett, M.V.T., N.F., C.R., T. Barton, J.C., J.H., U.W., H.D., J.J.C., A.R., E.P., N.D., S.L., T.P., R.M., F.S., D. Swales, A.S., D.P., R.K., L. Ladle, C.P., S.G.-P., P.P., R.D., C.S., J.W., A.M.J., A.V.R., R.S. and A.C.R. provided samples and archaeological contextualization. J.G., P.J., E.A., K.D., C.J.E., S.J., L. Lacher, S. Sabin, W.H., R.R., S. Brace, M.A.B., N.B., K.C., E.H., L.I., A.M.L., M.M., K. Stewardson and F.Z. performed the laboratory work. J.G., D. Sayer, P.J., M.P., S.S.E., K.H.S.M., N.P., L.P., D.R. and S. Schiffels analysed the genetic data. M.P., C.J.E., Å.J.V., N.R., B.K.-K., V.J.S., I.B., A.C., K.I.B., T. Booth, H.S., M.G.T., A.H., M.B.R., D.R., J.K. and S. Schiffels supervised the laboratory work, sampling or analysis. J.G., D. Sayer and S. Schiffels wrote the paper with input from all coauthors.

**Funding** Open access funding provided by Max Planck Society.

**Competing interests** The authors declare no competing interests.

**Additional information**
**Correspondence and requests for materials** should be addressed to Duncan Sayer or Stephan Schiffels.

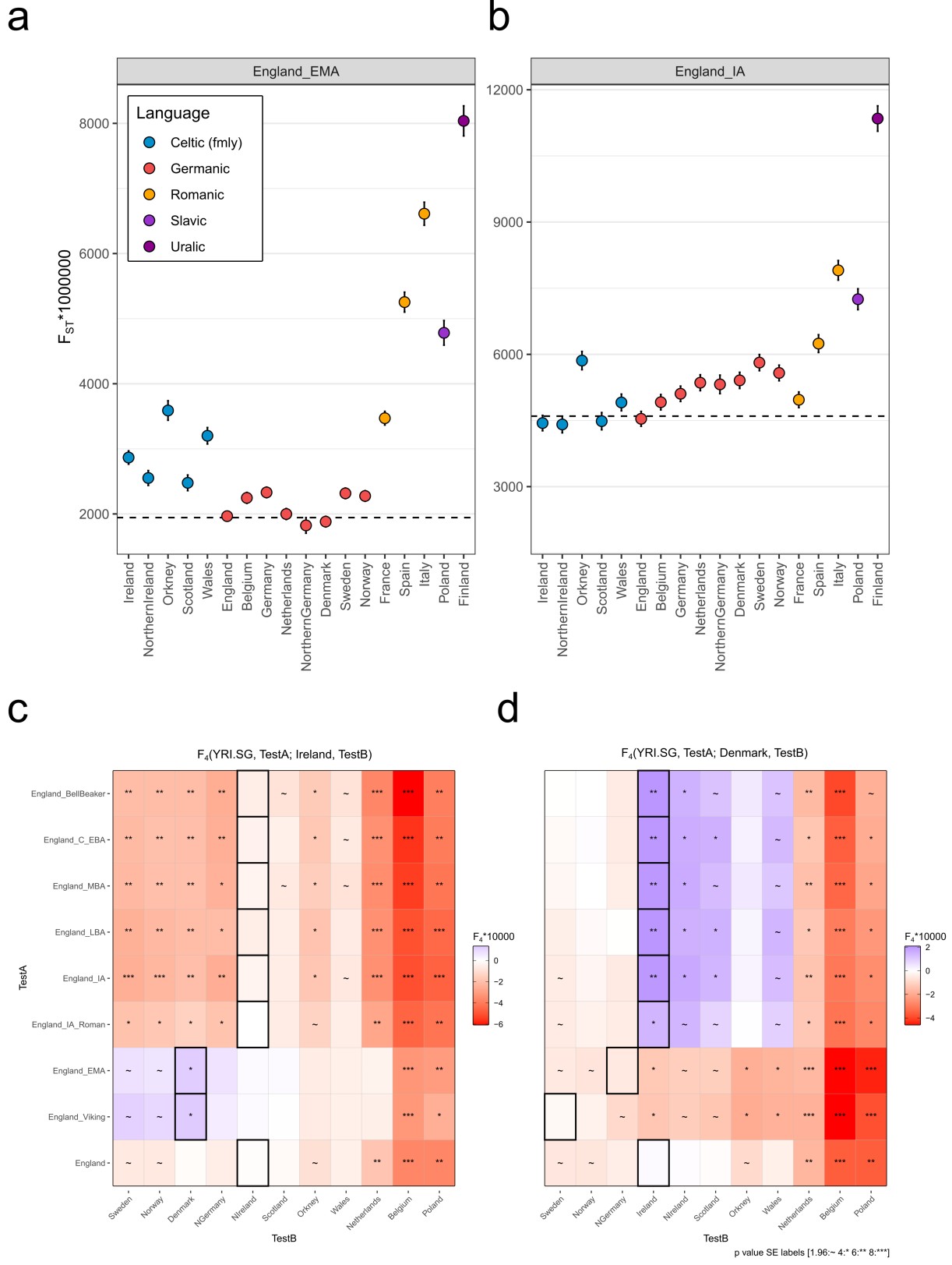

**Extended Data Fig. 1 | Genetic affinity statistics between ancient and present-day northwestern European populations.** a) $F_{ST}$ between relevant present-day Europeans and England_EMA (n = 285). Populations are coloured to their respective language family affiliation. Belgium is classified as Germanic, since Belgian samples were recruited mainly amongst patients from the northern Flemish-speaking region of Belgium. Error bars represent ± 3 standard errors. Samples sizes for present-day European populations are indicated in Supplementary Table 3.1 b) $F_{ST}$ between relevant present-day Europeans and England_IA (n = 290). Error bars represent ± 3 standard errors. c) F-statistics of the form $F_4$(YRI, TestA; Ireland, TestB). Negative values indicate that the test population is closer to Ireland than to TestB; positive values indicate that the test population is closer to TestB than to Ireland. d) Same for the F-statistics of the form $F_4$(YRI, TestA; Denmark, TestB).

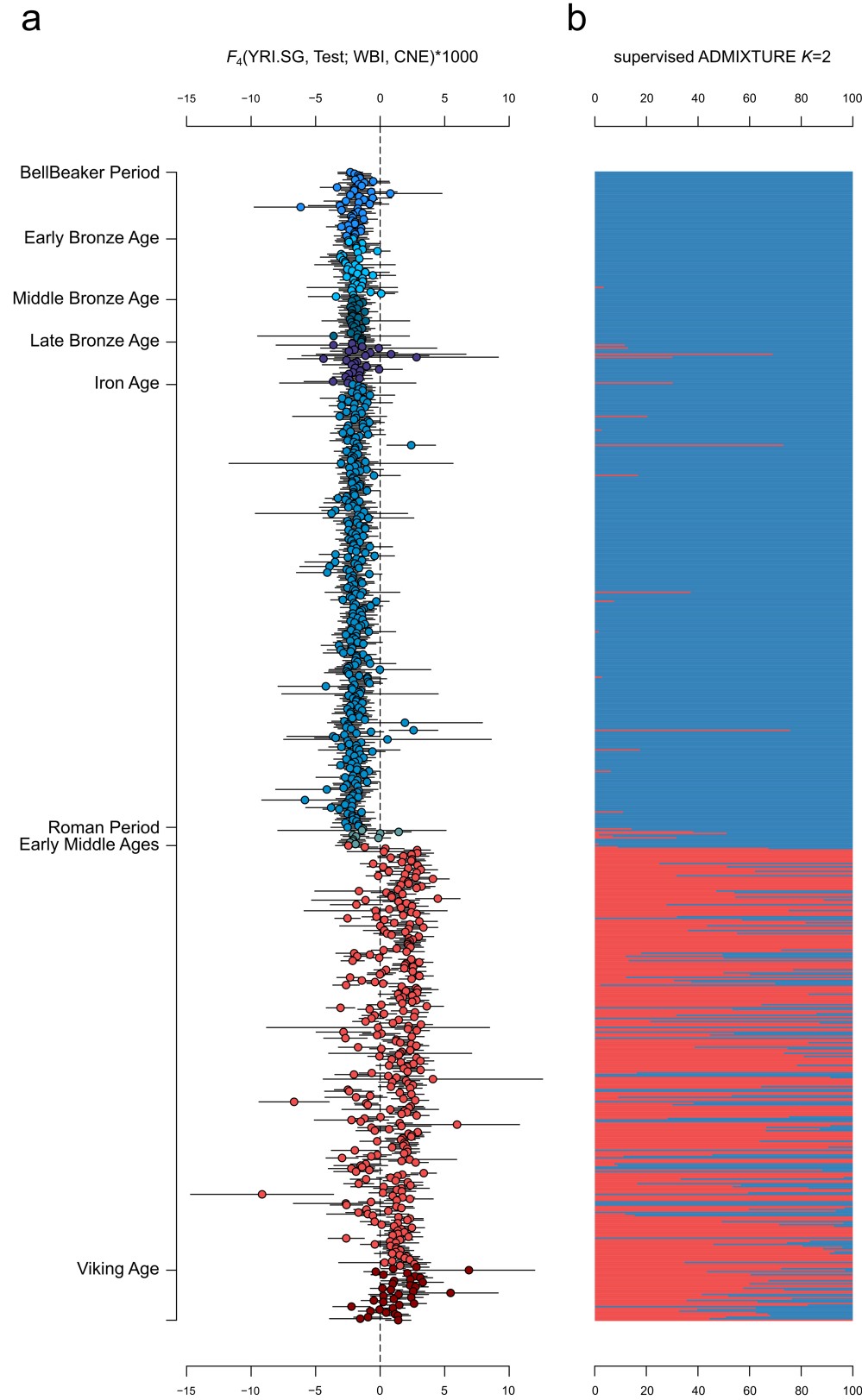

**Extended Data Fig. 2 | Individual-based ancestry decomposition and population affinities through time.** a) f-statistics of the form $F_4$(YRI, Test; WBI, CNE) for 758 ancient English individuals. Data are presented as point estimates for the respective $F_4$-statistic ± 2 standard errors. Negative values indicate that the test individual is closer to WBI than to CNE; positive values indicate that the test population is closer to CNE than to WBI. b) Modelling ancient post-Neolithic individuals from England and Ireland as a mixture of CNE individuals (red) and the WBI individuals (blue). Data are presented as admixture proportions. Each bar represents genome-wide mixture proportions for one individual. Individuals are ordered chronologically.

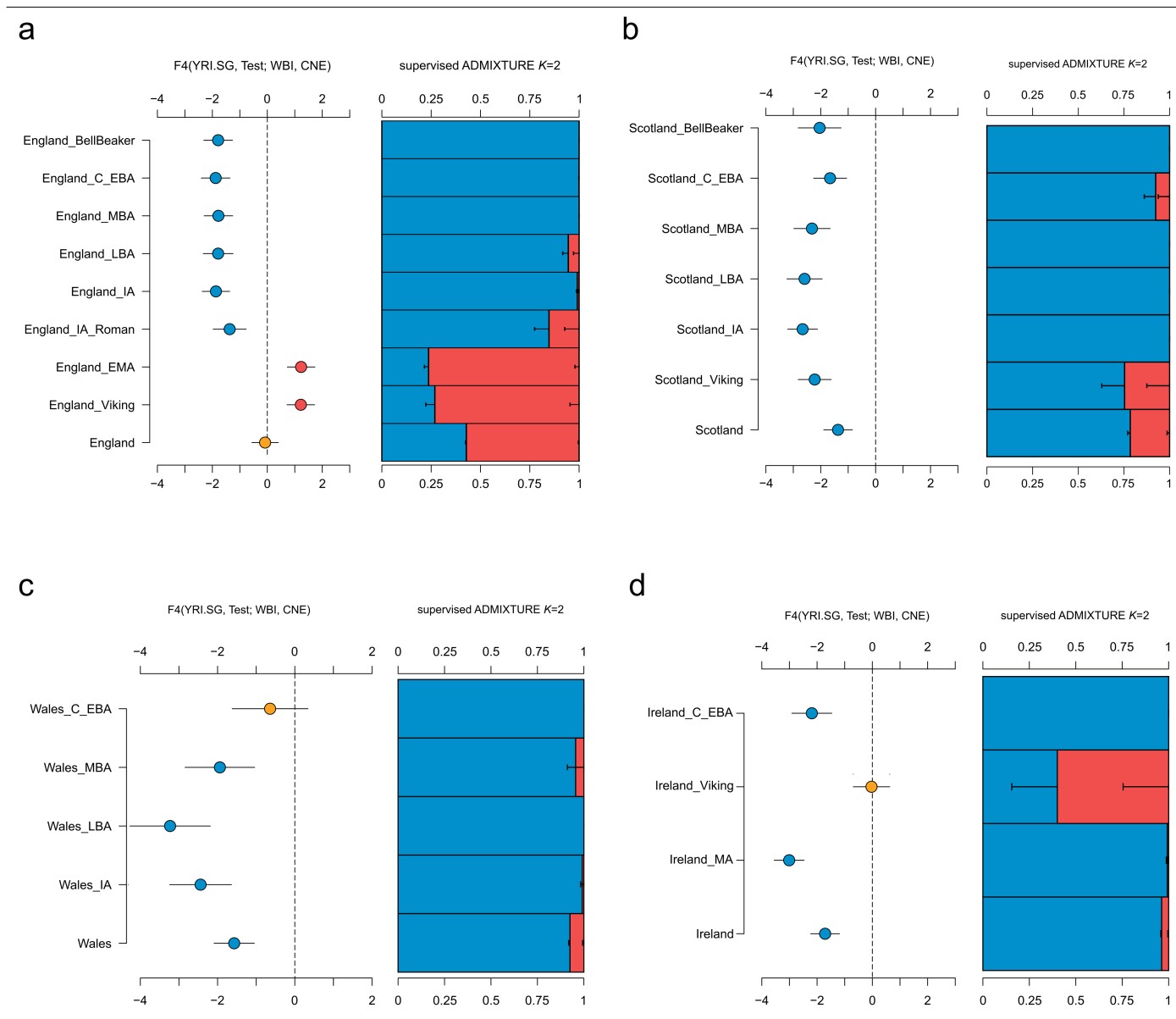

**Extended Data Fig. 3 | Regional ancestry decomposition and population affinities through time.** a) for England: left) f-statistics of the form $F_4$(YRI, Test; WBI, CNE). Data are presented as exact $F_4$-values. Negative values indicate that the test population is closer to WBI than to CNE; positive values indicate that the test population is closer to CNE than to WBI. Error bars represent ± 2 standard errors. Samples sizes for Test populations are indicated in Supplementary Table 3.3 right) Mean CNE and WBI ancestry proportions per period as inferred using supervised ADMIXTURE. Error bars represent ± 1 standard error of the mean. b) same for Scotland. c) same for Wales. d) same for Ireland.

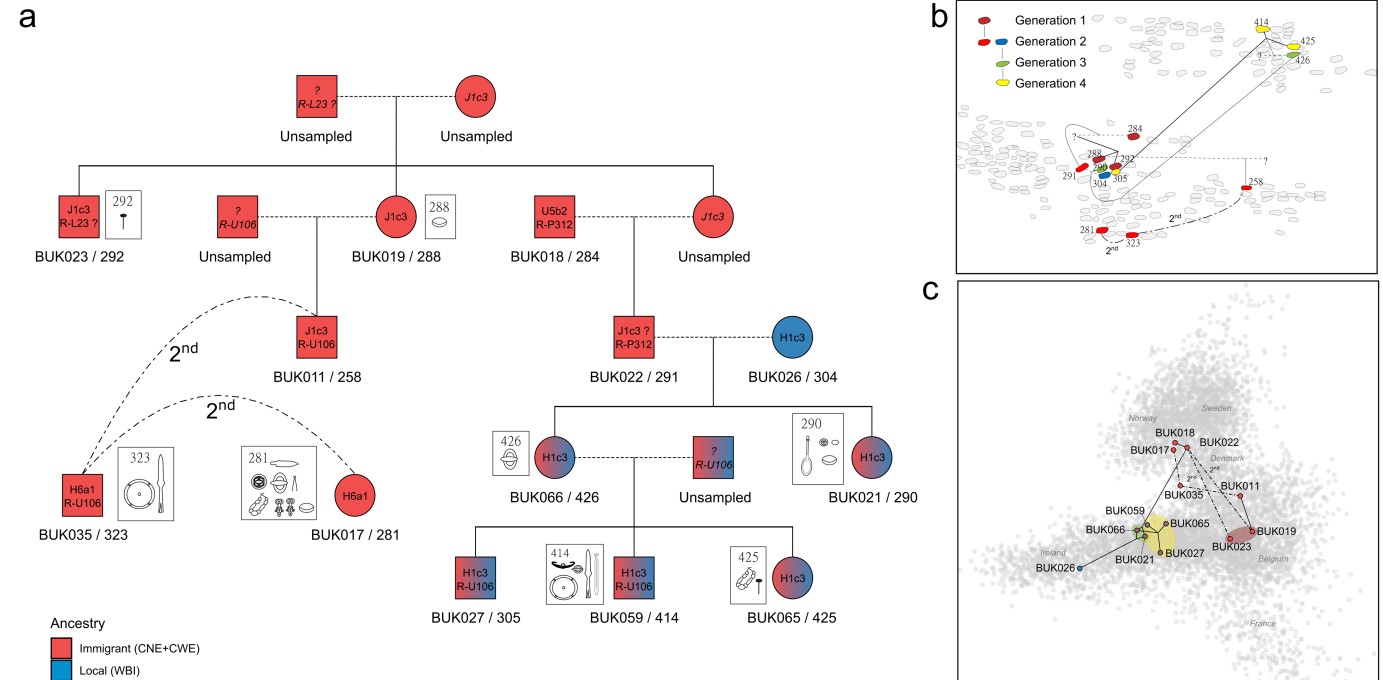

**Extended Data Fig. 4 | Family tree reconstruction featuring integration of local ancestry into an immigrant kin group.** a) The genetic pedigree of 13 related individuals at Dover Buckland. Indicated are the mtDNA haplogroups, Y-chromosome haplogroups, and associated grave goods of each individual. Males are depicted as squares, females as circles. b) Spatial distribution of the addressed burials across the site. Genetically related burials are connected with lines. c) Genetic distribution of the addressed individuals across a Principal Components Analysis of present-day genomes from northwestern Europe. Genetically related individuals are connected with lines.

a

b

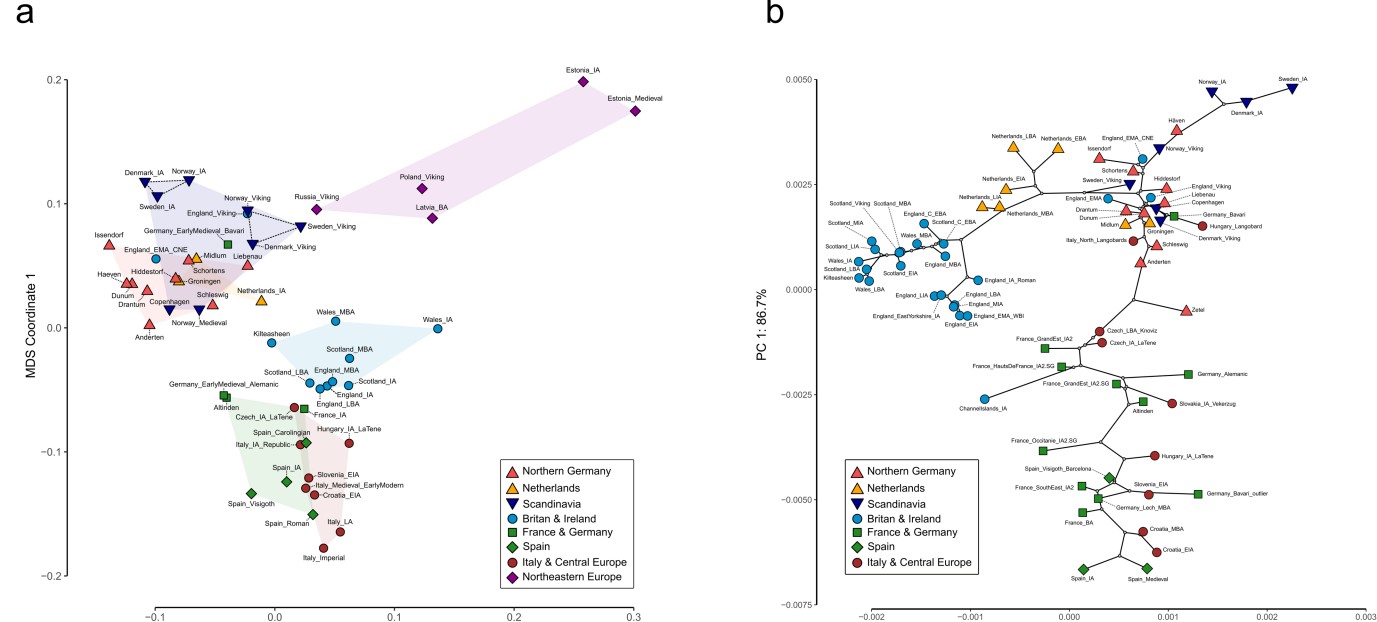

**Extended Data Fig. 5 | Visualisations of genetic affinity between early medieval English individuals and contemporary continental populations.** a) Multidimensional Scaling Plot of pairwise $F_3$ distances of the form $F_3$(CHB, ancient population A, ancient population B). b) PCA of $F_4$ statistics of the form

$F_4$(YRI, ancient population; TestA, TestB). TestA and TestB iterate through 15 present-day European populations (Methods). Additionally, a neighbour-joining tree of the same dataset was projected onto the two PCs. England_EMA_CNE are those early medieval English individuals who have exclusively CNE ancestry.

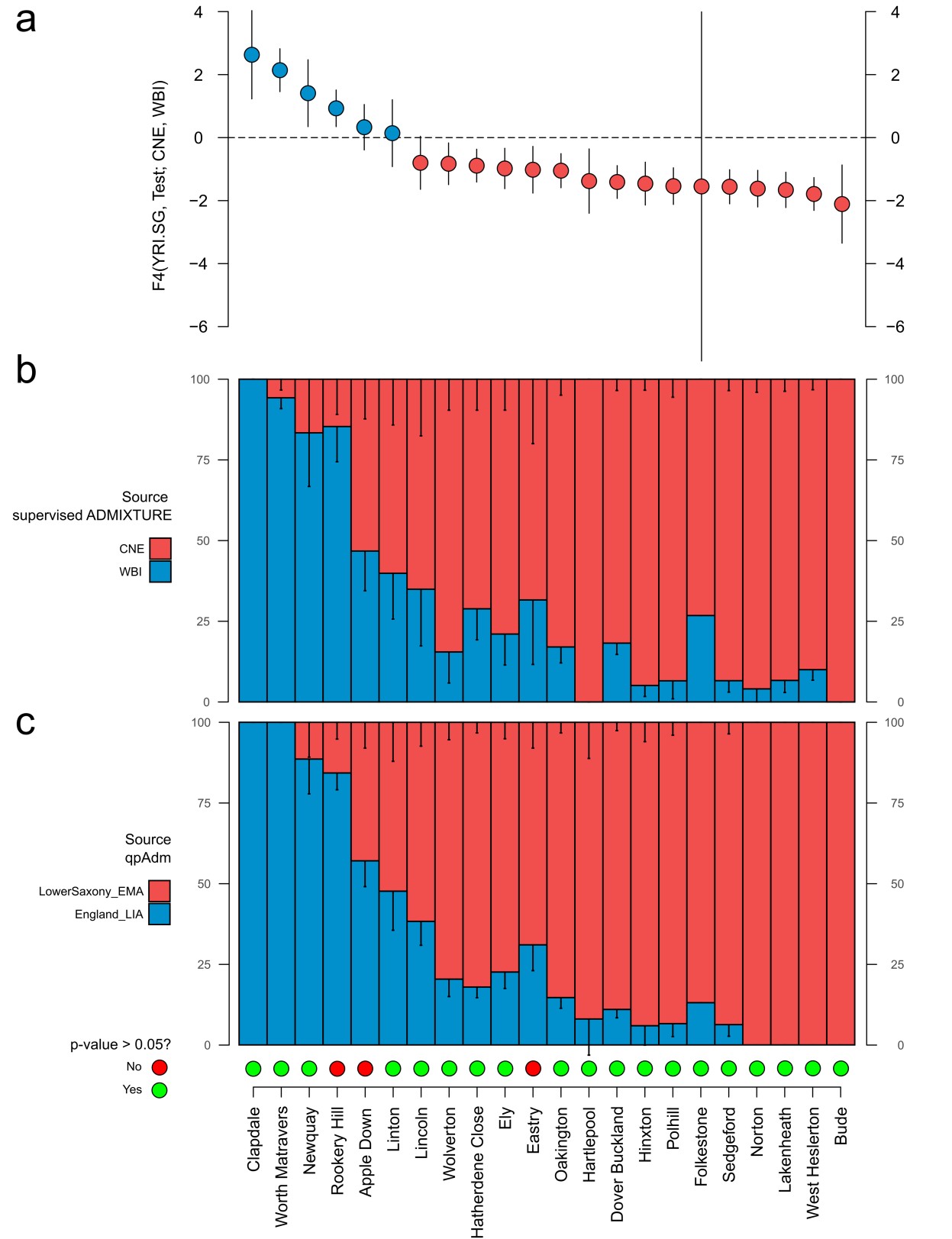

**Extended Data Fig. 6 | Measures of CNE ancestry in early medieval English sites.** a) $F_4$ statistics of the form $F_4$(YRI, Site; WBI, CNE). Data are presented as point estimates ± 2 standard errors. Samples sizes for early medieval English sites are indicated in Supplementary Table 5.3 b) Mean supervised ADMIXTURE proportions at K = 2. Error bars represent ± 1 standard error of the mean. c) qpAdm admixture proportions and p-values using a two-way admixture model of England_EMA with England_LIA_Roman (n = 32) and LowerSaxony_EMA (n = 39) as sources. Error bars represent ± 1 standard error.

a

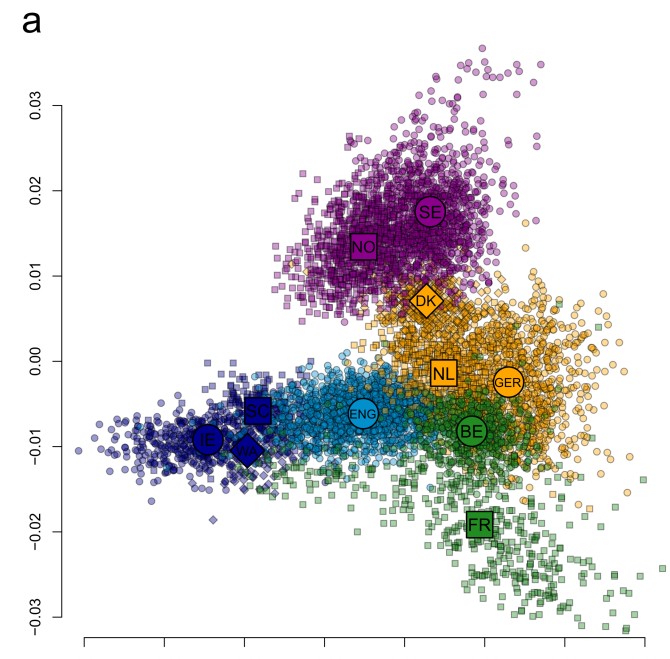

b

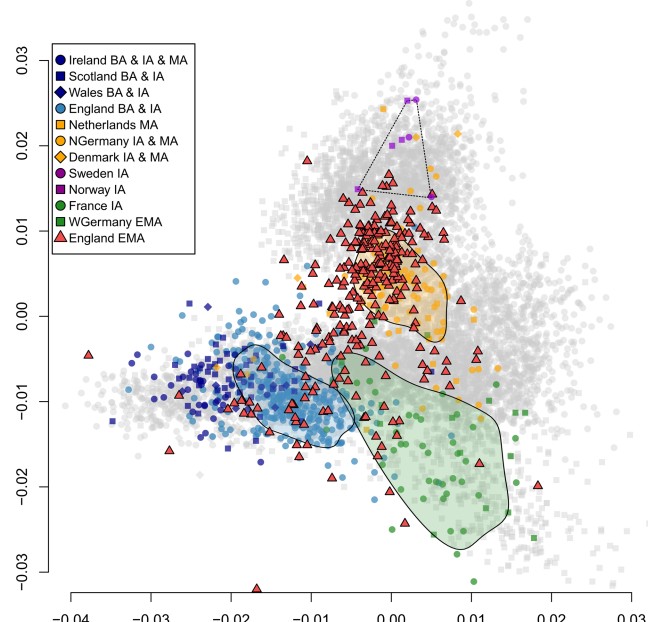

**Extended Data Fig. 7 | Extended Principal Components Analysis.** a) Principal Components Analysis of present-day genomes from northwestern Europe. IE = Northern Ireland & Ireland, WA = Wales, SC = Scotland, ENG = England, NL = Netherlands, GER = Northern Germany, DK = Denmark, NO = Norway, SE = Sweden, BE = Belgium, FR = France. b) Genetic structure of published and novel ancient individuals in this study, projected onto a). Polygons indicate where 2/3 of the data is located (England BA+IA, North Sea IA+EMA, and France IA+WGermany EMA, respectively). The Scandinavian Iron Age samples are connected with lines for clarity. Western Germany comprises samples from Alt-Inden; Northern Germany comprises samples from Lower Saxony, Mecklenburg-Vorpommern, and Schleswig-Holstein; Denmark_MA comprises samples from Copenhagen. BA = Bronze Age (EBA, MBA, and LBA = early, middle, and late BA), IA = Iron Age, RA = Roman Age, MA = Middle Ages (EMA=early MA), For rough time boundaries of the samples used here, see Methods.

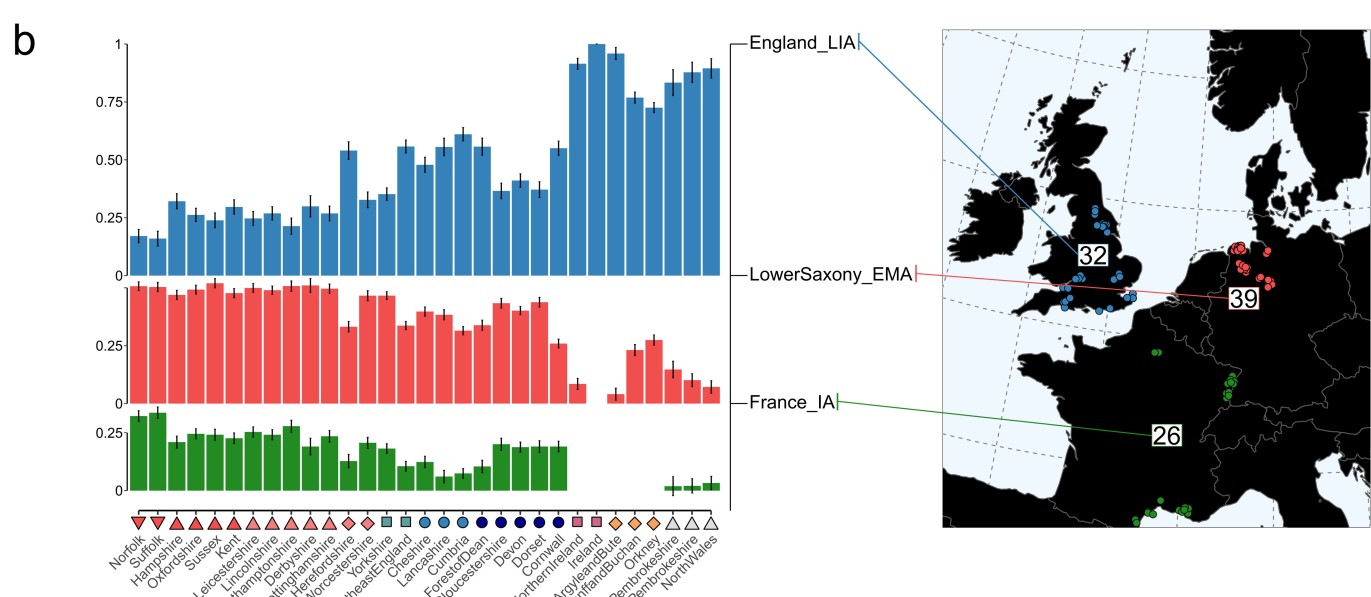

**Extended Data Fig. 8 | Correlations of early medieval ancestry with archaeological and linguistic evidence. a)** Distribution of early Anglo-Saxon cemeteries, Celtic river names, and Frankish objects. Ancestry proportions from supervised ADMIXTURE at K = 3 are shown for early Anglo-Saxon sites. CNE ancestry is shown in red, WBI in blue, and CWE in green. **b)** qpAdm ancestry proportions for 31 present-day populations from Britain and Ireland using LowerSaxony_EMA as a source for CNE ancestry. Number of individuals used for each source is indicated (England_LIA, LowerSaxony_EMA, and France_IA; n = 32, 39, and 26, respectively). Samples sizes for present-day PoBI sampling regions are indicated in Supplementary Table 5.16. Error bars represent ±1 standard error.

# Reporting Summary

## Statistics

For all statistical analyses, confirm that the following items are present in the figure legend, table legend, main text, or Methods section.

| n/a | Confirmed | |
|---|---|---|
| ☐ | ☒ | The exact sample size (*n*) for each experimental group/condition, given as a discrete number and unit of measurement |
| ☒ | ☐ | A statement on whether measurements were taken from distinct samples or whether the same sample was measured repeatedly |
| ☐ | ☒ | The statistical test(s) used AND whether they are one- or two-sided *Only common tests should be described solely by name; describe more complex techniques in the Methods section.* |
| ☒ | ☐ | A description of all covariates tested |
| ☐ | ☒ | A description of any assumptions or corrections, such as tests of normality and adjustment for multiple comparisons |
| ☐ | ☒ | A full description of the statistical parameters including central tendency (e.g. means) or other basic estimates (e.g. regression coefficient) AND variation (e.g. standard deviation) or associated estimates of uncertainty (e.g. confidence intervals) |
| ☐ | ☒ | For null hypothesis testing, the test statistic (e.g. *F*, *t*, *r*) with confidence intervals, effect sizes, degrees of freedom and *P* value noted *Give P values as exact values whenever suitable.* |
| ☐ | ☒ | For Bayesian analysis, information on the choice of priors and Markov chain Monte Carlo settings |
| ☐ | ☒ | For hierarchical and complex designs, identification of the appropriate level for tests and full reporting of outcomes |
| ☐ | ☒ | Estimates of effect sizes (e.g. Cohen's *d*, Pearson's *r*), indicating how they were calculated |

*Our web collection on statistics for biologists contains articles on many of the points above.*

## Software and code

Policy information about availability of computer code

| Data collection | No specific software was used for sample collection. Genotype data were generated from sequencing reads. All software used in this study is listed below. |
|---|---|
| Data analysis | All software used in this work is publicly available. Corresponding publications are cited in the main text and supplementary material. List of software and respective versions: AdapterRemoval (v2.3.1), Burrows-Wheeler Aligner (v0.7.12), DeDup (v0.12.2), mapDamage (v2.0.6), BamUtil (v1.0.14), EAGER (v1), Sex.DetERRmine (v1.1.2) (https://github.com/TCLamnidis/Sex.DetERRmine), ANGSD (v0.915), Schmutzi (v1.5.4), contamMix (v1.0-12), PMDtools (v0.50), pileupCaller (v1.4.0.2), samtools (v1.3.1), Geneious R9.8.1, HaploGrep 2 (v2.4.0), READ (https://bitbucket.org/tguenther/read) (vf541d55), lcMLkin (https://github.com/COMBINE-lab/maximum-likelihood-relatedness-estimation) (v0.5.0), PLINK (v1.90b3.29), Picard tools (v2.27.3), ADNA-Tools (https://github.com/DReichLab/ADNA-Tools) (v3b4357d), smartpca (v16000; EIGENSOFT v6.0.1), qp3Pop (v.435; ADMIXTOOLS v3.0), qpDstat (v.755; ADMIXTOOLS v3.0), Treemix (v1.12), qpWave (v410), qpAdm (v.810), LOCATOR (v1.2), ADMIXTURE (v1.3). The code used in Supplementary Note 2 ("Estimating sex-biased ancestry from uniparental markers in the presence of variable admixture proportions") can be found at https://github.com/stschiff/AngloSaxon_Y-chrom_sex-bias. Data visualisation and descriptive statistical tests were performed in R (v4.1.1). The following R packages were used: Rsamtools (v2.12.0), binom (v1.1-1.1), ape (v.5.6-2), phytools (v1.0-3), psych (v2.2.5), vegan (v2.6-2), factoextra (v1.0.7), ggplot2 (v3.3.6), ggExtra (v0.10.0), ggforce (v0.3.3), rnaturalearth (v0.1.0), sf (v1.0.-8), raster (v3.5-21), elevatr (v0.4.2), rgdal (v1.5-32), spatstat (v2.3-4), maptools (v1.1-4), gstat (v2.0-9), sp (v1.5-0), labdsv (v2.0-1), igraph (v1.3.4), magrittr (v2.0.3), dplyr (v1.0.9), reshape 2 (v1.4.4), and tidyverse (v.1.3.2). Y-chromosome and mtDNA haplogroups were determined using the ISOGG SNP index (v15.73) and PhyloTree (v17-FU1) reference databases, respectively. |

For manuscripts utilizing custom algorithms or software that are central to the research but not yet described in published literature, software must be made available to editors and reviewers. We strongly encourage code deposition in a community repository (e.g. GitHub). See the Nature Portfolio guidelines for submitting code & software for further information.

## Data

Policy information about availability of data

All manuscripts must include a data availability statement. This statement should provide the following information, where applicable:
- Accession codes, unique identifiers, or web links for publicly available datasets
- A description of any restrictions on data availability
- For clinical datasets or third party data, please ensure that the statement adheres to our policy

Raw sequence data (bam files) from the 479 newly reported ancient individuals will be available prior publication from the European Nucleotide Archive under accession number PRJEB54899. Published genotype data for the present-day British sample are available from the WTCCC via the European Genotype Archive (https://www.ebi.ac.uk/ega/) under accession number EGAD00010000634. Published genotype data for the present-day Irish sample are available from the WTCCC via the European Genotype Archive under accession number EGAD00010000124. Published genotype data for the rest of the present-day European samples are available from the WTCCC via the European Genotype Archive under accession number EGAD00000000120. Published genotype data for the Dutch samples are available by the GoNL request process from The Genome of the Netherlands Data Access Committee (DAC) (https://www.nlgenome.nl). The Genome Reference Consortium Human Build 37 (GRCh37) is available via the National Center for Biotechnology Information under accession number PRJNA31257. The revised Cambridge reference sequence is available via the National Center for Biotechnology Information under NCBI Reference Sequence NC_012920.1. Previous published genotype data for ancient individuals was reported by the Reich Lab in the Allen Ancient DNA Resource v.50.0 (https://reich.hms.harvard.edu/allen-ancient-dna-resource-aadr-downloadable-genotypes-present-day-and-ancient-dna-data).

# Field-specific reporting

Please select the one below that is the best fit for your research. If you are not sure, read the appropriate sections before making your selection.

☒ Life sciences   ☐ Behavioural & social sciences   ☐ Ecological, evolutionary & environmental sciences

For a reference copy of the document with all sections, see nature.com/documents/nr-reporting-summary-flat.pdf

# Life sciences study design

All studies must disclose on these points even when the disclosure is negative.

| | |
|---|---|
| Sample size | We did not rely on statistical methods to predetermine sample sizes. Sample sizes for ancient populations depended solely on the availability of archaeological material and on ancient DNA preservation. |
| Data exclusions | We selected 379 samples for genome-wide analyses, out of all (n=494) sequenced samples, based on their endogenous content and low contamination estimates. The exclusion criteria mentioned above were pre-established. For analyses including present-day Europeans, individuals with noticeable non-European ancestry were excluded, as described in the Supplementary (Supplementary Note 2; Reference data). Furthermore, closely related individuals were excluded from analyses requiring population allele frequencies. |
| Replication | We studied unique entities (past and present populations) and did not perform experiments or study various treatments, so replication is not applicable. But we note that samples from the same population carry similar genetic signatures. Moreover, genome-wide data allows for the analysis of multiple realisations of the sample history, by studying hundreds of thousands of SNP sites. |
| Randomization | We studied unique entities (past and present populations) and did not perform experiments or study various treatments, so randomization is not applicable. |
| Blinding | We studied unique entities (past and present populations) and did not perform experiments or study various treatments, so blinding is not applicable. |

# Reporting for specific materials, systems and methods

We require information from authors about some types of materials, experimental systems and methods used in many studies. Here, indicate whether each material, system or method listed is relevant to your study. If you are not sure if a list item applies to your research, read the appropriate section before selecting a response.

## Materials & experimental systems

| n/a | Involved in the study |
|---|---|
| ☒ | ☐ Antibodies |
| ☒ | ☐ Eukaryotic cell lines |
| ☐ | ☒ Palaeontology and archaeology |
| ☒ | ☐ Animals and other organisms |
| ☒ | ☐ Human research participants |
| ☒ | ☐ Clinical data |
| ☒ | ☐ Dual use research of concern |

## Methods

| n/a | Involved in the study |
|---|---|
| ☒ | ☐ ChIP-seq |
| ☒ | ☐ Flow cytometry |
| ☒ | ☐ MRI-based neuroimaging |

# Palaeontology and Archaeology

**Specimen provenance**

Provenance of all samples is outlined in Supplementary Section 1. In summary:

Samples from Apple Down were provided by the Novium Museum, Chichester; Permission to analyze was granted to coauthor Ceiridwen Edwards in 2012.

Samples from Dover Buckland were provided by the Canterbury Archaeological Trust. Permission to analyse was granted to coauthor Duncan Sayer in 2019.

Samples from Widemouth Bay and Newquay, Crantock were provided by the Royal Cornwall Museum. Permission to analyse was granted to coauthor Tom Booth in 2019.

Samples from Fox Holes Cave were provided by the Tom Lord's collection at Lower Winskill Farm. Permission to analyse was granted to coauthor Tom Booth in 2015.

Samples from Oakington, Polhill and Eastry Updown were provided by the University of Central Lancashire. Permission to analyse was granted to coauthor Duncan Sayer in 2017.

Samples from Ely and Hatherdene Close were provided by Oxford Archaeology East. Permission to analyse was granted to coauthor Duncan Sayer in 2017.

Samples from Hartlepool, Norton Bishopsmill and Norton East Mill were provided by Tees Archaeology. Permission to analyse was granted to coauthor Tom Booth in 2019.

Samples from Dover Hill were provided by the Folkestone Museum. Permission to analyse was granted to coauthor Tom Booth in 2016.

Samples from RAF Lakenheath where provided by Cotswold Archaeology Suffolk. Permission to analyse was granted to coauthor Duncan Sayer in 2017.

Samples from Lincoln Castle where provided by FAS Heritage. Permission to analyse was granted to coauthor Tom Booth in 2015.

Samples from Rookery Hill were provided by the Brighton Museum and Art Gallery. Permission to analyse was granted to coauthor Tom Booth in 2019.

Samples from Sedgeford were provided by the Sedgeford Historical and Archaeological Research Project (SHARP). Permission to analyse was granted to coauthor Stephan Schiffels in 2017.

Samples from West Heslerton were provided by the Landscape Research Centre. Permission to analyse was granted to coauthors Martin Richards and Ceiridwen Edwards in 2016.

Samples from Wolverton were provided by the Milton Keynes Museum. Permission to analyse was granted to coauthor Tom Booth in 2015.

Samples from Worth Matravers where provided by the East Dorset Antiquarian Society. Permission to analyse was granted to coauthors Martin Richards and Ceiridwen Edwards in 2012.

Samples from Kilteasheen were provided by the Institute of Technology Sligo. Permission to analyse was granted to coauthor Kirsten Bos in 2014.

Samples from Groningen were provided by the Gemeente Groningen. Permission to analyse was granted to coauthor Eveline Altena in 2017.

Samples from Midlum were provided by the Noordelijk Archeologisch Depot. Permission to analyse was granted to coauthor Eveline Altena in 2017.

Samples from Copenhagen were provided by the Museum of Copenhagen. Permission to analyse was granted to coauthor Hannes Schroeder in 2019.

Samples from Alt-Inden were provided by the LVR-Amt für Bodendenkmalpflege im Rheinland. Permission to analyse was granted to coauthor Stephan Schiffels in 2021.

Samples from Hannover-Anderten, Liebenau, Hiddestorf and Issendorf were provided by the Landesmuseum Hannover. Permission to analyse was granted to coauthor Stephan Schiffels in 2020.

Samples from Dunum were provided by the Niedersächsisches Institut für historische Küstenforschung (NIhK). Permission to analyse was granted to coauthor Stephan Schiffels in 2020.

Samples from Drantum, Schortens and Zetel were provided by the Landesmuseum Natur und Mensch Oldenburg. Permission to analyse was granted to coauthor Stephan Schiffels in 2020.

Samples from Häven were provided by the University of Rostock. Permission to analyse was granted to coauthor Joscha Gretzinger in 2021

Samples from Schleswig were provided by the Stiftung Schleswig-Holsteinische Landesmuseen Schloss Gottorf. Permission to analyse was granted to coauthor Ben Krause-Kyora in 2017.

**Specimen deposition**

Specimens were returned to the owning institutions after laboratory analyses.

**Dating methods**

A subset of bone samples were sent to the Curt-Engelhorn-Center Archaeometry gGmbH (CECA) to be dated by 14C. Collagen was extracted from the bone samples (modified Longin method), purified by ultrafiltration (fraction >30kD) and freeze-dried. The analysis was performed on a MICADAS-type AMS system at CECA. The isotopic ratios 14C/12C and 13C/12C of samples, calibration standard (Oxalic Acid-II), blanks and control standards were measured simultaneously in the AMS. 14C-ages are normalized to delta 13C=-25‰ and calibrated using the dataset IntCal20 and software SwissCal (L.Wacker, ETH-Zürich).

☒ Tick this box to confirm that the raw and calibrated dates are available in the paper or in Supplementary Information.

**Ethics oversight**

No ethical oversight was required strictly. However, we confirm that all analyses followed established ethical guidelines for archaeogenetic research, as detailed in Wagner et al., AJHG, 2020 and Alpaslan-Roodenberg, Nature, 2021.

Note that full information on the approval of the study protocol must also be provided in the manuscript.