## [Peer Review File · Nature]

Manuscript Title: The Anglo-Saxon migration and the formation of the Early English gene pool

Reviewer Comments & Author Rebuttals

Reviewer Reports on the Initial Version:

Referees' comments:

Referee #1 (Remarks to the Author):

This paper focuses on a much-storied transition in the British past, the post Roman influx of Anglo-Saxons. It presents a substantial new ancient genome data set both from the British Isles and relevant continental regions and features a thorough range of analyses. It establishes substantial migration into Britain in the early middle ages.

The migration of 'Germanic' peoples in this event has been addressed by genomics before, notably by the People of the British Isles publication in Nature (Leslie et al) and also directly by a several ancient genomics studies (including one by the corresponding author). Conclusions here largely agree with these prior works (it has a feature of the field that studies with low sample numbers have usually been confirmed by larger subsequent works), and these are acknowledged in the paper.

However, this study has interesting novelty in that:

- It allows an unambiguous assertion that migration was substantial, and had wide effect. It allows rejection that culture and language were transmitted by a small elite influx. I think this is important because of the (ideological) opposition to migration as an explanatory factor for cultural transition which lingers within academic communities studying the human past.
- it also shows a separate migratory influence from French Iron Age populations which has a southern English impact
- it contains credible analyses estimating the continental origins of migrating peoples
- it has detail on the interaction between the different ancestral components, geography and status (adduced from material culture). The patterns are not simple and defy generalisation, but are interesting and well-discussed.
- the ancient data allows an assessment of the lasting impact of these migrations on present day populations.
- there is a substantial and useful supplement with additional analyses.

Anglo-Saxon migrations are a primary component of the origins mythology of English people and language and therefore are of wide general interest. This paper is the most comprehensive assessment of these to date and does make a substantial step change in the level of knowledge around these. Therefore I think this work is publishable in Nature.

Some points to be addressed follow.

The paper uses both 1240k SNP capture data and low coverage shotgun sequenced genomes. There are known batch effects that can lead to different behaviours of these data in some of the analyses used in the paper. I think some consideration/discussion of these are necessary

The spatio-temporal ancient genome sampling in England is excellent, there is a very good Irish sample, but there are no data points in either Scotland or Wales. This is a weakness and could be discussed - however, I do not recommend that further sampling should be required for acceptance.

Figure 2 could do with some cosmetic work. Why not keep some correspondence between colours and geography in the two plots? It might be easier in the reader to include some in-diagram labels

Figure 5 includes plots of CNE (ie Germanic region) ancestry and French Iron Age-like ancestry in modern British Isles populations. These maps are an important communication, given the importance that the general public may attach to these ancestries. As such I wonder are they too simplistic or smoothed. It is mentioned that they use 32 population estimates but the geographical origins of these are not shown on the maps. Are these points sufficiently dispersed that they capture the variegated ancestries of Scotland, Wales and Ireland? For example, it has been known since blood group analyses in the 1960s that Ireland has substantial English ancestry in both the Southeast and North East from very large historical migrations (the supplement mentions an increased CNE fraction in a Northern Irish sample). Also, is the higher CNE in Aberdeen an artefact or a signal of something real in NE Scotland? How much are the Welsh and Northern English contours driven by proximity to Ireland and Scotland? At least the distributions of the underlying data from which interpolation is achieved should be shown - and perhaps limitations of resolution mentioned.

Minor point: line 372, more context (location) of Worth Matravers needs to be mentioned Line 383, is it correct to refer to the French IA influx as a "gene flow event", implying a discrete happening?

There is a tradition of discussion of Cornwall-Brittany affinity, based on placenames and language. How does the France IA - South Britain migration fit with this?

Referee #2 (Remarks to the Author):

This paper presents the results of the first large-scale, genome-wide aDNA analysis of early medieval burials in NW Europe. The study substantially expands the number of individuals analysed in this way, above all for England. The focus on England relates to a longstanding debate about the extent to which the radical changes in material culture and language seen in post-Roman Britain, especially

in the South and East, were due to large-scale immigration from the NW European mainland as described in early written sources, or to 'elite dominance' by a small number of warbands combined with a large-scale assimilation of the Britons. The results are thus both original and significant.

In addition to 205 individuals from England, 181 individuals from neighbouring European regions were analysed. Ideally, more individuals from Northern Gaul, i.e. the Low Countries and northern France, would have been included (only two Dutch cemeteries were examined). The lack of analyses from this region may well have been due to issues around preservation and access, but the reasons for this omission should be mentioned, especially in light of the conclusions drawn in lines 375-81, indicating a French 'strand' to the three-way population model. More should also be said about the rationale for the selection of particular cemeteries; the nature of the 'comprehensive time-transect of sites' (line 115) for England also needs to be explained.

The ancient genomes were 'projected onto' present-day genetic variation amongst NW Europeans. Variable amounts of NW European ancestry are indicated for individuals from early medieval England, with some mixing between western British/Irish samples and samples from the NW European mainland. The authors then estimate the genetic contribution to ancestry composition using ADMIXTURE. The results, which indicate relatively low but increasing 'Continental northern Europe' (CNE) ancestry through the Bronze Age to Roman periods, are plausible, though based on small numbers of analysed individuals. The majority of early medieval individuals analysed from England, however, derive most or all of their ancestry from CNE, with an average of around 74%. This dominance of CNE ancestry is substantially reduced for southern and SW England and for Ireland, as would be expected. The variability revealed within the same cemetery and seemingly even within family or household groups is particularly significant (lines 204-13). The lack of a clear link between ancestry and burial rite (i.e. richly furnished versus unfurnished/'poor' burials) is also a significant finding. This finding appears to be in tension, however, with Lines 226-39, where it is argued that at individual cemeteries, e.g. Hatherdene Close and Apple Down, links can be seen between burial rite, burial location and ancestry. This is stated, however without any supporting evidence such as a cemetery plan showing the location of particular burials, or statistical analysis that would allow the strength of these apparent patterns to be evaluated – e.g. how many burials are involved? Supporting evidence is likewise needed in relation to lines 240-46, where a 'systematic distinction' is claimed between cemeteries which show (apparently) 'integration' of CNE and British/Irish groups, and those where an immigrant group is 'dominant'. This is quite a big claim, without any evidence provided in support (Extended Fig. 3a does not present the evidence on which these statements are based).

The paper presents important new data and the conclusions are original. The identification of NW German individuals as having a particularly close relationship to the individuals from England is important, if unsurprising; the link with Denmark is more unexpected, although all of the Danish samples are relatively late, i.e. 11th- 12th century. This should be emphasised more clearly (and earlier) in the paper, which only mentions it in passing in lines 285-7. These results do not have an obvious bearing on the questions regarding post-Roman Britain; indeed, it is unclear why these later burials and those from the medieval town of Schleswig were included. While it is true that earlier inhumations are scarce in this region, it may have been better to omit these burials entirely rather than muddy the water with results from a very different historical context.

Lines 404-14 should be more clearly expressed. What exactly is being argued here and how are these observations new? Continuing mobility across the N Sea from the Late Roman period to the Viking Age and beyond has not seriously been in doubt for the past 30 years. It is the scale and impact of the migration that continues to be debated.

In conclusion, I recommend the paper for publication, subject to the issues raised above being addressed. In addition, listed below are a number of comparatively minor matters for correction, primarily relating to the introductory section. The subject matter will be of wide interest, not only to archaeologists, but also to medieval historians and demographers. The paper will be much cited.

Minor Corrections:

The introductory, scene-setting section (c lines 1-100) is overly compressed/telescoped and will be hard for those without a background in this period to follow.

The authors sometimes refer to the study as examining 'medieval' burials, sometimes 'Early medieval'. They should be consistent. 'Early medieval' would be better.

The colours used on the inset, Fig. 2 are difficult to distinguish (e.g. between Netherlands and Ireland), despite the different symbols, even when enlarged.

Line 48: specify 'in the fifth century' after 'administration'

Line 71: As noted later in the paper, there are few close links between England and the 'Jutland peninsula' in terms of material culture, and certainly little or no sign of 'Jutes'. Instead, the close links with the Frisian coast and Low countries should be noted here.

Line 73: Quoit brooches (or, more correctly, Quoit brooch style metalwork) is thought to have been made in Britain.

Line 75: southern Scandinavian origins

Line 81 Instead of 'Romanised British population', 'Romano-British population'

The cemetery of Eastry, Updown is consistently misspelled 'Upton'.

Line 92 Should be: 'preferred theoretical position of many archaeologists from...'

Line 93-4: preferred a model of 'elite dominance involving small, mobile warbands, and the acculturation of the indigenous British population.'

96 Why 're-population'?

Line 106: delete 'was inferred to'

Line 436: is migration inferred, or demonstrated?

Referee #3 (Remarks to the Author):

Review manuscript # 2021-12-19567

Gretzinger et al. Present manuscript entitled "The Anglo-Saxon migration and the formation of the Early English gene pool". The authors present genomic data from 368 medieval individuals from north-western Europe (including 197 genomes from England) and compare the data with thousands of published ancient and modern North European genomes in order to shed light on the source and

character of Anglo-Saxon migrations as well as population structure of ancient and modern populations of the British Isles and elsewhere. Though the topic has been previously discussed in numerous archaeological and genetic publications (as pointed out by authors themselves), it has not been thus far approached in such scale. Therefore, the publication should be of interest to the general public. The authors mostly use a combination of various well-established and commonly used tools in ancient DNA analyses (especially combination of Principal Component Analysis, ADMIXTURE, qpAdm, qpWave and various F statistics) and assess validity of the results through calculation of correlation coefficients and providing p-values for presented analyses. The undertaken statistical analyses are appropriate, and well described, while any weaknesses are also described and discussed. Something that should be applauded. By running multiple carefully designed testes with various meticulously curated reference datasets (both modern and ancient) the authors make a compelling case supporting their claims. The abstract, introduction and conclusions are clear, while the multilayer analyses performed and described in the supplement in detail are skillfully condensed in the main text to be used in the discussion in combination with and considering the archaeological evidence. The manuscript is elegant and solid.

That being said, I must point out that the originality regarding the topic, some of the main conclusions (e.g. Lower Saxony being the main source of Anglo-Saxon migrations or the distribution of ancestry clines observed in the British Isles) as well as the analytical tools applied is somewhat limited considering the usually ground-breaking character of research published by Nature. Nevertheless, the presented paper, its well-supported findings, the multitude of newly generated genomes and the analytical craftsmanship deserve publication and will be of great value to the scientific community.

Here are some minor points which the authors may or may not find useful:

Main text:

Could you please clarify how many samples were analyzed? While the difference between the number of samples extracted and those used in genomic analyses is understandable, the number of genomes newly reported varied across the manuscript, as the paper lists “studied” - 368, “sampled” - 386; 395 in Table S1 and 396 bams submitted to ENA and in the methods part, 395 again.

When modelling ancient English populations as mixture of English_IA (English Iron Age and Roman samples pooled together) and LowerSaxony_EMA are you not concerned about the difference in the number of individuals in those two groups (n=8 and n=41 respectively)?

While you make it quite clear in the supplement that mtDNA haplogroup variation could not be used to support claims of genetic turnover, it could be useful to place that information in the main text since there is a whole paragraph describing Y chromosome haplogroups. A single sentence would suffice and provide balanced reference to both uniparental markers.

Please check Figure 2 caption, as it seems somewhat confusing. Also, would it not be logical to have the present-day individuals from France and Belgium added in the main PCA seeing WE ancestry component is important in the discussion?

In caption to Figure 3 adding (n=) numbers of individuals representing different periods, i.e.: EBA, Roman and EMA could be useful while comparing panels in fig 3b.

Please check sentence three in section “Tracing ancestry sources from across the North Sea”.

Supplementary file:

Archaeology

As I am not an archaeologist it is impossible for me to judge or comment on the appropriateness of the selected archaeological materials. The sheer number of samples analyzed is impressive and the archaeological descriptions are quite informative, albeit with rather numerous punctuation mistakes, ambiguities, case inconsistencies and typos in the section describing the English sites (e.g. “aged 4045 years”, p3; „whith his head”, p3; “with as Swanton”, p6; „hands resented on her shoulders”, p6; “the bone is c.20m shorter”, p13; „Grave 77 (Sample ID I20665_d) was an adult SEX and was found flexed”, p18; etc.).

There are some minor inconsistencies regarding placing sample IDs in the supplement, e.g. some sites are supplied with sample IDs (very useful), while others are not. Adding the samples IDs everywhere would make it easier for the reader to navigate the manuscript.

Kinship

While most relations described seem well resolved, if one should like to further investigate some the unresolved relations using ngsRelate (Korneliussen & Moltke, 2015; Hanghøj et al., 2015) to obtain X chromosomal kinship coefficients could prove quite useful.

Supplementary Note 4 - Identifying outliers – in table S26 at least five individuals present significantly higher affinity to either Sweden or Norway, while having over 250k SNPs overlapping with the dataset. Are those considered “low coverage”?

Supplementary Figure 3.2. It would be very helpful if you could add component legend with colors and names to each of the panels.

Supplementary Figure 3.7. c. Possibly using K=5 values with components described as in Supplementary Figure 3.2. could be clearer.

Supplementary Figure 4.2. Please check if the caption is correct.

Supplementary Figure 5.2. The positioning of Alt-Inden with respect to WBI and CNE was quite striking and could be highlighted earlier in the text.

Supplementary Figure 5.10. Seeing that those presented are eastern connections it would be really useful to have anything from Poland included in the test as well. There might not be suitable medieval or Iron Age data available, but there should Bronze Age data published.

Supplementary Figure 6.9. Adding legends with the color intensity/component proportion to plots

would be useful as the plots seem almost identical.

Excel supplementary:

Table S23 – Should one add Z scores here?

Table S24 – The caption lists CHB.SG not YRI as the outgroup population.

Table S25 – Seeing that none of the tested individuals is statistically closer to NOR than CNE, might it be useful to have the f_4 or values of a form $F_4(\text{YRI.SG, Test; NOR, WBI})$ here as well, especially that S27 tests CNE+NOR+WBI mixture? Despite this being irrelevant information as to the question of identifying potential affinities associated with Saxons, Angles, and Jutes it would be interesting to have that information at hand.

Table S39 – Should the results for Eastry be marked as ‘infeasible’ as well?

Table S59 – Are the average F_{ST} distances correct here? They seem odd. Are those values *100000?
Please check.

References:

Korneliussen, T.S., Moltke, I., 2015. NgsRelate: a software tool for estimating pairwise relatedness from next-generation sequencing data. *Bioinformatics* 31, 4009–4011.
<https://doi.org/10.1093/bioinformatics/btv509>

Hanghøj, K., Moltke, I., Andersen, P.A., Manica, A., Korneliussen, T.S., 2019. Fast and accurate relatedness estimation from high-throughput sequencing data in the presence of inbreeding. *GigaScience* 8. <https://doi.org/10.1093/gigascience/giz034>

Author Rebuttals to Initial Comments:

Referee comments are in *black italics*, our response in blue non-italics.

Referee #1

This paper focuses on a much-storied transition in the British past, the post Roman influx of Anglo-Saxons. It presents a substantial new ancient genome data set both from the British Isles and relevant continental regions and features a thorough range of analyses. It establishes substantial migration into Britain in the early middle ages.

The migration of 'Germanic' peoples in this event has been addressed by genomics before, notably by the People of the British Isles publication in Nature (Leslie et al) and also directly by a several ancient genomics studies (including one by the corresponding author). Conclusions here largely agree with these prior works (it has a feature of the field that studies with low sample numbers have usually been confirmed by larger subsequent works), and these are acknowledged in the paper.

However, this study has interesting novelty in that:

- It allows an unambiguous assertion that migration was substantial, and had wide effect. It allows rejection that culture and language were transmitted by a small elite influx. I think this is important because of the (ideological) opposition to migration as an explanatory factor for cultural transition which lingers within academic communities studying the human past.

- it also shows a separate migratory influence from French Iron Age populations which has a southern English impact

- it contains credible analyses estimating the continental origins of migrating peoples

- it has detail on the interaction between the different ancestral components, geography and status (adduced from material culture). The patterns are not simple and defy generalisation, but are interesting and well-discussed.

- the ancient data allows an assessment of the lasting impact of these migrations on present day populations.

- there is a substantial and useful supplement with additional analyses.

Anglo-Saxon migrations are a primary component of the origins mythology of English people and language and therefore are of wide general interest. This paper is the most comprehensive assessment of these to date and does make a substantial step change in the level of knowledge around these. Therefore I think this work is publishable in Nature.

Some points to be addressed follow.

The paper uses both 1240k SNP capture data and low coverage shotgun sequenced genomes. There are known batch effects that can lead to different behaviours of these data

in some of the analyses used in the paper. I think some consideration/discussion of these are necessary

Response: Yes, there are indeed batch effects, and they potentially impact analyses to a varying degree.

First, with respect to PCA and ADMIXTURE analyses, we exclusively use those methods to *project* ancient samples onto present-day genetic variation. This means that principal components and latent ancestry sources are exclusively computed from present-day data, which does not exhibit any such batch effects, and therefore these projections are also not affected by them.

There is a potential issue with respect to ancestry decomposition using the *qpAdm* method. These are performed only in the analyses described in Supplementary Notes 3, 5 and 6, and displayed in Figure 5 of the main text. Within these analyses, we have undertaken measures to make sure our results are robust against batch effects. Specifically, when modeling present-day English groups as being admixed from ancient sources (*England_EMA_CNE*, *England_LIA*, *France_IA*), we have run three versions of each model: a) Both Shotgun and 1240K capture individuals are used in the sources, b) only Shotgun are used in the sources, c) only 1240K individuals are used in the sources. All differences in ancestry proportions between these runs are within the estimated standard errors, so we are confident that batch effects in these models are small (Supplementary Figure 5.20). In addition, we have also within this analysis ensured robustness against variable sample sizes in the different sources (Supplementary Figure 5.20).

When modeling ancient English samples as being admixed from ancient sources (Supplementary Section 3 and 5), we did not implement any additional control, in part because these models result in relatively large standard errors, which arguably are beyond the rather subtle batch effects. However, we did check that modeling estimates from supervised ADMIXTURE, F-Statistics, various *qpAdm* models and PCA position are strongly correlated with each other (e.g. Supplementary Figures 3.5, 5.5 and 5.19), which adds to our confidence that our estimates are robust and unbiased across different methods, ancient vs. modern sources, sample sizes or shotgun vs. capture.

The spatio-temporal ancient genome sampling in England is excellent, there is a very good Irish sample, but there are no data points in either Scotland or Wales. This is a weakness and could be discussed - however, I do not recommend that further sampling should be required for acceptance.

Response: Indeed, our coverage does not include all of Great Britain, and particularly Scotland and Wales are not covered currently. We acknowledge this now more clearly in the Discussion: "Specifically, in early medieval western England, Wales and Scotland, as well as more generally in England during the Norman period, further aDNA sampling may clarify how CNE ancestry spread and was subsequently diluted. "

Figure 2 could do with some cosmetic work. Why not keep some correspondence between colours and geography in the two plots? It might be easier in the reader to include some in-diagram labels.

Response: Figure 2 has changed in various aspects, including indeed a now more aligned color-scheme between modern and ancients, as suggested.

Figure 5 includes plots of CNE (ie Germanic region) ancestry and French Iron Age-like ancestry in modern British Isles populations. These maps are an important communication, given the importance that the general public may attach to these ancestries. As such I wonder are they too simplistic or smoothed. It is mentioned that they use 32 population estimates but the geographical origins of these are not shown on the maps. Are these points sufficiently dispersed that they capture the variegated ancestries of Scotland, Wales and Ireland? For example, it has been known since blood group analyses in the 1960s that Ireland has substantial English ancestry in both the Southeast and North East from very large historical migrations (the supplement mentions an increased CNE fraction in a Northern Irish sample). Also, is the higher CNE in Aberdeen an artefact or a signal of something real in NE Scotland? How much are the Welsh and Northern English contours driven by proximity to Ireland and Scotland? At least the distributions of the underlying data from which interpolation is achieved should be shown - and perhaps limitations of resolution mentioned.

Response: We agree that interpolation maps can lead to false impressions in places with sparse data. However, in this case we believe that the sampling density is high enough to justify these maps. To make this more transparent, we have now included the anchor points for these interpolations as black dots into the two map figures in Figure 5. The signal in Wales is indeed based on local measures from the region, and no spill-over effect from Ireland, as now becomes clear with the dots.

Note that we do provide also barcharts and pie charts of these proportions per region, for example see Extended Data Figure 8, Supplementary Figure 5.21, as well as our new analyses behind Supplementary Figure 5.23.

*Minor point: line 372, more context (location) of Worth Matravers needs to be mentioned
Line 383, is it correct to refer to the French IA influx as a “gene flow event”, implying a discrete happening?*

Response: We have added “[Worth Matravers], at the southern coast of Dorset” to give location context. We have changed “gene flow events” into “gene flow processes” in this sentence.

There is a tradition of discussion of Cornwall-Brittany affinity, based on placenames and language. How does the France IA - South Britain migration fit with this?

Response: We lack the data to systematically address this. There is a single sample from Brittany that we now include in our analyses, but that does not yield any generalisable insights into this connection. We believe our general signal of *France_IA* ancestry in England is too strong to be explained or substantially affected by this regionally-specific connection. The bulk of samples used here for modeling this connection are widely distributed across France and England. More specific sampling would be necessary to look into this question.

Referee #2

This paper presents the results of the first large-scale, genome-wide aDNA analysis of early medieval burials in NW Europe. The study substantially expands the number of individuals analysed in this way, above all for England. The focus on England relates to a longstanding debate about the extent to which the radical changes in material culture and language seen in post-Roman Britain, especially in the South and East, were due to large-scale immigration from the NW European mainland as described in early written sources, or to 'elite dominance' by a small number of warbands combined with a large-scale assimilation of the Britons. The results are thus both original and significant.

In addition to 205 individuals from England, 181 individuals from neighbouring European regions were analysed. Ideally, more individuals from Northern Gaul, i.e. the Low Countries and northern France, would have been included (only two Dutch cemeteries were examined). The lack of analyses from this region may well have been due to issues around preservation and access, but the reasons for this omission should be mentioned, especially in light of the conclusions drawn in lines 375-81, indicating a French 'strand' to the three-way population model. More should also be said about the rationale for the selection of particular cemeteries; the nature of the 'comprehensive time-transect of sites' (line 115) for England also needs to be explained.

Response: The scope of this study is England and continental Northern Europe. Naturally there are limits in scope to every study, but we agree that further sampling in particular in France, which we did not have an opportunity for, would have been useful. However, since the last submission, a major new study (Patterson et al. 2022, Nature) appeared, which increased the Iron Age sample size in France by 29, which together with the previous 19 individuals from Brunel et al. 2020 now provides a comfortable basis for using France as a direct source in our modeling. Regarding Dutch cemeteries, we note that cremation was a common burial practice in the early medieval period in continental Northern Europe, which largely limits opportunities for aDNA in this region. Note that the Patterson et al. study also included Bronze and Iron Age data from the Netherlands, so our data situation has improved there as well since the last submission. On the last point: We now clarify further "We target a comprehensive time-transect of sites in the south and east of England, spanning predominantly the time period 450 - 850 CE"

The ancient genomes were 'projected onto' present-day genetic variation amongst NW Europeans. Variable amounts of NW European ancestry are indicated for individuals from

early medieval England, with some mixing between western British/Irish samples and samples from the NW European mainland. The authors then estimate the genetic contribution to ancestry composition using ADMIXTURE. The results, which indicate relatively low but increasing ‘Continental northern Europe’ (CNE) ancestry through the Bronze Age to Roman periods, are plausible, though based on small numbers of analysed individuals. The majority of early medieval individuals analysed from England, however, derive most or all of their ancestry from CNE, with an average of around 74%. This dominance of CNE ancestry is substantially reduced for southern and SW England and for Ireland, as would be expected. The variability revealed within the same cemetery and seemingly even within family or household groups is particularly significant (lines 204-13).

Response: Note that our coverage of Bronze- and Iron-Age sites has substantially increased in the revised version due to incorporating new data from Patterson et al. 2021.

The lack of a clear link between ancestry and burial rite (i.e. richly furnished versus unfurnished/‘poor’ burials) is also a significant finding. This finding appears to be in tension, however, with Lines 226-39, where it is argued that at individual cemeteries, e.g. Hatherdene Close and Apple Down, links can be seen between burial rite, burial location and ancestry. This is stated, however without any supporting evidence such as a cemetery plan showing the location of particular burials, or statistical analysis that would allow the strength of these apparent patterns to be evaluated – e.g. how many burials are involved? Supporting evidence is likewise needed in relation to lines 240-46, where a ‘systematic distinction’ is claimed between cemeteries which show (apparently) ‘integration’ of CNE and British/Irish groups, and those where an immigrant group is ‘dominant’. This is quite a big claim, without any evidence provided in support (Extended Fig. 3a does not present the evidence on which these statements are based).

Response: Yes, we agree and thank the referee for bringing this up. We now include a large stack of statistical analyses that systematically test for relationships between grave goods, ancestry and other variables (see Supplementary Note 7), not the least made possible due to 74 new samples from Dover Buckland that had not been included in the first submission. We believe this has improved the study substantially and added additional insights, as we now mention in the abstract and the Results section. A highlight of this analysis is the finding of a significant correlation between CNE ancestry and the presence of grave goods in particular in women, as we now clearly report in the text. Note that site maps are in fact available for most sites now (see Supplementary Note 1).

The paper presents important new data and the conclusions are original. The identification of NW German individuals as having a particularly close relationship to the individuals from England is important, if unsurprising; the link with Denmark is more unexpected, although all of the Danish samples are relatively late, i.e. 11th- 12th century. This should be emphasised more clearly (and earlier) in the paper, which only mentions it in passing in lines 285-7. These results do not have an obvious bearing on the questions regarding post-Roman Britain; indeed, it is unclear why these later burials and those from the medieval town of

Schleswig were included. While it is true that earlier inhumations are scarce in this region, it may have been better to omit these burials entirely rather than muddy the water with results from a very different historical context.

Response: Comparative analyses with ancient Denmark are only performed in the Results section titled "Tracing ancestry sources from across the North Sea", which is exactly where we mention this potential caveat, and we don't see how it would be relevant earlier in the text. We also do not want to over-emphasise this point, since our analysis actually shows that the gene pool in Denmark was remarkably stable between Late Bronze Age, Iron Age and medieval period. This can be seen in Supplementary Figure 3.2 (ADMIXTURE profiles), Supplementary Figure 4.2 (affinities using F4 statistics) and Supplementary Figure 4.4 (clustering on ancestry components).

We agree that our own contribution to the Danish sample set is indeed rather late (the Copenhagen data). However, by far the biggest part of sampling around the Baltic Sea is provided by the previous Viking study (Margaryan et al. 2020, Nature), which spans the entire first millennium CE (ca. 0 - 1100 CE). We therefore think that our newly reported later samples from Copenhagen and Schleswig hardly muddy the water in that respect, even though they do indeed stand out temporally from the rest of the continental and English samples that we provide.

Lines 404-14 should be more clearly expressed. What exactly is being argued here and how are these observations new? Continuing mobility across the N Sea from the Late Roman period to the Viking Age and beyond has not seriously been in doubt for the past 30 years. It is the scale and impact of the migration that continues to be debated.

Response: We added Sedgeford to the mentioned sentences, as one of the key sites that affect the temporal aspect of this reasoning. We do believe that our evidence for the temporal scope of these migrations is noteworthy and extends the current knowledge, so deserves to be discussed. The migration period is often considered to be bounded in the 5-7th centuries whereas we argue that our data pushes that into the 7th and 8th century or middle Saxon period in a significant way, effectively joining the Saxon and Viking Migrations as continuous.

In conclusion, I recommend the paper for publication, subject to the issues raised above being addressed. In addition, listed below are a number of comparatively minor matters for correction, primarily relating to the introductory section. The subject matter will be of wide interest, not only to archaeologists, but also to medieval historians and demographers. The paper will be much cited.

Minor Corrections:

The introductory, scene-setting section (c lines 1-100) is overly compressed/telescoped and will be hard for those without a background in this period to follow.

Response: We went through this introductory part again and tried to clarify the language.

The authors sometimes refer to the study as examining 'medieval' burials, sometimes 'Early medieval'. They should be consistent. 'Early medieval' would be better.

Response: We agree and went through the entire text to add "early" where appropriate (typically when England is referred to, but not on the continent, where our time-span is larger).

The colours used on the inset, Fig. 2 are difficult to distinguish (e.g. between Netherlands and Ireland), despite the different symbols, even when enlarged.

Response: Figure 2 has changed, and this point, among others, has been addressed.

Line 48: specify 'in the fifth century' after 'administration'

Response: The sentence now reads (slightly changed): "The end of Roman administration in 5th century Britain preceded a dramatic transformation in material culture, architecture, manufacturing and agricultural practice, and was accompanied by language change."

Line 71: As noted later in the paper, there are few close links between England and the 'Jutland peninsula' in terms of material culture, and certainly little or no sign of 'Jutes'. Instead, the close links with the Frisian coast and Low countries should be noted here.

Response: There are material culture links to the Jutland peninsula. We have taken up the suggestion to note the Frisian coast in the first paragraph of the introduction.

Line 73: Quoit brooches (or, more correctly, Quoit brooch style metalwork) is thought to have been made in Britain.

Response: We removed this object class from the sentence.

Line 75: southern Scandinavian origins

Response: We added "southern"

*Line 81 Instead of 'Romanised British population', 'Romano-British population'
The cemetery of Eastry, Updown is consistently misspelled 'Upton'.*

Response: We changed to "Romano-British" and fixed spelling mistakes, thanks for catching those.

Line 92 Should be: 'preferred theoretical position of many archaeologists from...'

Response: Changed as suggested.

Line 93-4: preferred a model of 'elite dominance involving small, mobile warbands, and the acculturation of the indigenous British population.'

Response: Changed as suggested, but changed "indigenous" to "local".

96 Why 're-population'?

Response: This sentence has changed and the term isn't anymore there.

Line 106: delete 'was inferred to'

Response: Indeed, removed.

Line 436: is migration inferred, or demonstrated?

Response: We have opted for "found" in this place now.

Referee #3

Review manuscript # 2021-12-19567

Gretzinger et al. Present manuscript entitled "The Anglo-Saxon migration and the formation of the Early English gene pool". The authors present genomic data from 368 medieval individuals from north-western Europe (including 197 genomes from England) and compare the data with thousands of published ancient and modern North European genomes in order to shed light on the source and character of Anglo-Saxon migrations as well as population structure of ancient and modern populations of the British Isles and elsewhere. Though the topic has been previously discussed in numerous archaeological and genetic publications (as pointed out by authors themselves), it has not been thus far approached in such scale. Therefore, the publication should be of interest to the general public. The authors mostly use a combination of various well-established and commonly used tools in ancient DNA analyses (especially combination of Principal Component Analysis, ADMIXTURE, qpAdm, qpWave and various F statistics) and assess validity of the results through calculation of correlation coefficients and providing p-values for presented analyses. The undertaken statistical analyses are appropriate, and well described, while any weaknesses are also described and discussed. Something that should be applauded. By running multiple carefully designed testes with various meticulously curated reference datasets (both modern and ancient) the authors

make a compelling case supporting their claims. The abstract, introduction and conclusions are clear, while the multilayer analyses performed and described in the supplement in detail are skillfully condensed in the main text to be used in the discussion in combination with and considering the archaeological evidence. The manuscript is elegant and solid.

That being said, I must point out that the originality regarding the topic, some of the main conclusions (e.g. Lower Saxony being the main source of Anglo-Saxon migrations or the distribution of ancestry clines observed in the British Isles) as well as the analytical tools applied is somewhat limited considering the usually ground-breaking character of research published by Nature. Nevertheless, the presented paper, its well-supported findings, the multitude of newly generated genomes and the analytical craftsmanship deserve publication and will be of great value to the scientific community.

Here are some minor points which the authors may or may not find useful:

Main text:

Could you please clarify how many samples were analyzed? While the difference between the number of samples extracted and those used in genomic analyses is understandable, the number of genomes newly reported varied across the manuscript, as the paper lists “studied” - 368, “sampled” - 386; 395 in Table S1 and 396 bams submitted to ENA and in the methods part, 395 again.

Response: These discrepancies are due to the multiple layers of analyses performed. Some samples have failed, or were flagged “QUESTIONABLE”, meaning that they have too shallow nuclear genomic data allowing for sex-determination but not more. Some of these have mitochondrial DNA analyses, but no nuclear DNA was sufficiently preserved. We have made this clear now in the first Results section: “We sampled skeletal remains from 494 ancient northwestern Europeans [...]. After quality filtering (Methods), 460 genome-wide individuals were available for analysis. [...]”. In the abstract, we now mention 460 samples as being “studied”. The numbers are now also in sync with Table S1.

When modelling ancient English populations as mixture of English_IA (English Iron Age and Roman samples pooled together) and LowerSaxony_EMA are you not concerned about the difference in the number of individuals in those two groups (n=8 and n=41 respectively)?

Response: In our re-analysis of our data in conjunction with the newly added Patterson et al. data, these sample sizes are much more similar (33 vs 39). We note that in several analyses of qpAdm, we have run multiple versions of the analyses with shotgun vs. capture selections, and also with downsampled groups to check robustness against sample size differences. See also our first response to referee #1.

While you make it quite clear in the supplement that mtDNA haplogroup variation could not be used to support claims of genetic turnover, it could be useful to place that information in the main text since there is a whole paragraph describing Y chromosome haplogroups. A single sentence would suffice and provide balanced reference to both uniparental markers.

Response: We have improved our analysis of the mtDNA, and now find that they do actually show signals of population turnover mirroring the effects measured from our genome-wide analyses. See Supplementary Note 2. In the main text we now write “Similarly, mitochondrial genomes show evidence of female lineage population turnover from North-Sea bordering regions (Supplementary Note 2).”

Please check Figure 2 caption, as it seems somewhat confusing. Also, would it not be logical to have the present-day individuals from France and Belgium added in the main PCA seeing WE ancestry component is important in the discussion?

Response: Thanks for catching the grammar error in the caption text. Fixed. Regarding France and Belgium, we have discussed this among the authors quite at length, and concluded that at this point in the manuscript, we prefer a visualization of the data that highlights the dominant population structure emerging *within the early medieval English* group, rather than highlight the (mostly) later changes that have shaped present-day England. But in response to this comment, we have elevated a Supplementary Figure showing the more comprehensive PCA now to Extended Data Figure 7, to bring the PCA with France and Belgium closer to the reader's attention.

In caption to Figure 3 adding (n=) numbers of individuals representing different periods, i.e.: EBA, Roman and EMA could be useful while comparing panels in fig 3b.

Response: Good suggestion, we have added this now.

Please check sentence three in section “Tracing ancestry sources from across the North Sea”.

Response: This part has changed now and hopefully is clearer (and typo-free).

Supplementary file:

Archaeology

As I am not an archaeologist it is impossible for me to judge or comment on the appropriateness of the selected archaeological materials. The sheer number of samples analyzed is impressive and the archaeological descriptions are quite informative, albeit with rather numerous punctuation mistakes, ambiguities, case inconsistencies and typos in the section describing the English sites (e.g. “aged 4045 years”, p3; „whith his head”, p3; “with as Swanton”, p6; „hands resented on her shoulders”, p6; “the bone is c.20m shorter”, p13; „Grave 77 (Sample ID I20665_d) was an adult SEX and was found flexed”, p18; etc.).

Reponse: Indeed embarrassing, we have gone through this again, thanks for flagging! Note that many of those listings have now disappeared from the Supplement, as we now list this information in Supplementary Tables 1 and 7.1.

There are some minor inconsistencies regarding placing sample IDs in the supplement, e.g. some sites are supplied with sample IDs (very useful), while others are not. Adding the samples IDs everywhere would make it easier for the reader to navigate the manuscript.

Response: Indeed, we have gone through and improved consistency, see also the reply to the previous comment. Note that the information situation for the sites is heterogeneous, with some being described in publications, others not at all.

Kinship

While most relations described seem well resolved, if one should like to further investigate some the unresolved relations using ngsRelate (Korneliussen & Moltke, 2015; Hanghøj et al., 2015) to obtain X chromosomal kinship coefficients could prove quite useful.

Response: We thank the referee for this suggestion, but decided to not follow this up further, for now. We are in fact also preparing more targeted publications on some of the more densely sampled cemeteries, such as Dover Buckland.

Supplementary Note 4 - Identifying outliers – in table S26 at least five individuals present significantly higher affinity to either Sweden or Norway, while having over 250k SNPs overlapping with the dataset. Are those considered “low coverage”?

Response: This paragraph was extensively edited (see Supp. Note 6). The statement was removed.

Supplementary Figure 3.2. It would be very helpful if you could add component legend with colors and names to each of the panels.

Response: This refers to the unsupervised admixture analyses, where latent ancestry sources (“components”) do not have any prior labels. Rather, their nature becomes clear through where they occur. We have improved clarity by splitting that plot into a “legend” part (Supplementary Figure 3.1) and a “sample” part (Supplementary Figure 3.2).

Supplementary Figure 3.7. c. Possibly using $K=5$ values with components described as in Supplementary Figure 3.2. could be clearer.

Response: Indeed. We have taken up this suggestion, but opted for $K=6$, to highlight the difference between Scandinavian and Northern German ancestry. Note that this Figure is now Supplementary Figure 3.8.

Supplementary Figure 4.2. Please check if the caption is correct.

Response: Thanks for catching. This figure has been replaced by a new plot (Supplementary Figure 6.2).

Supplementary Figure 5.2. The positioning of Alt-Inden with respect to WBI and CNE was quite striking and could be highlighted earlier in the text.

Response: We agree, but don't see scope for this in the main text, and feel it is appropriately noted in Supplementary section 4 (new numbering).

Supplementary Figure 5.10. Seeing that those presented are eastern connections it would be really useful to have anything from Poland included in the test as well. There might not be suitable medieval or Iron Age data available, but there should Bronze Age data published.

Response: This figure has been replaced by Table S4.6, and we have indeed added Polish Bronze Age data to the analysis.

Supplementary Figure 6.9. Adding legends with the color intensity/component proportion to plots would be useful as the plots seem almost identical.

Response: Added

Excel supplementary:

Table S23 – Should one add Z scores here? - Added

Table S24 – The caption lists CHB.SG not YRI as the outgroup population. - This is intentional.

Table S25 – Seeing that none of the tested individuals is statistically closer to NOR than CNE, might it be useful to have the f_4 or values of a form $F_4(\text{YRI.SG, Test; NOR, WBI})$ here as well, especially that S27 tests CNE+NOR+WBI mixture? Despite this being irrelevant information as to the question of identifying potential affinities associated with Saxons, Angles, and Jutes it would be interesting to have that information at hand. - In the context of already daunting numbers of tables and pages of supplement, we believe this test would yield too little additional information on top of the two existing forms $F_4(\text{YRI.SG, Test; CNE, WBI})$ and $F_4(\text{YRI.SG, Test; CNE, NOR})$, in particular given the close genetic relationship between NOR and CNE, and therefore the expected similarity of the patterns between $F_4(\text{YRI.SG, Test; CNE, WBI})$ and $F_4(\text{YRI.SG, Test; NOR, WBI})$.

Table S39 – Should the results for Eastry be marked as 'infeasible' as well? - This table has been removed and the information subsumed by Table S5.3 and S5.4. The result has changed due to new data in the sources and different sources.

*Table S59 – Are the average F_{ST} distances correct here? They seem odd. Are those values *100000? Please check. - Yes, it should be scaled, indeed. Fixed.*

References:

Korneliussen, T.S., Moltke, I., 2015. NgsRelate: a software tool for estimating pairwise relatedness from next-generation sequencing data. *Bioinformatics* 31, 4009–4011. <https://doi.org/10.1093/bioinformatics/btv509>

Hanghøj, K., Moltke, I., Andersen, P.A., Manica, A., Korneliussen, T.S., 2019. Fast and accurate relatedness estimation from high-throughput sequencing data in the presence of inbreeding. *GigaScience* 8. <https://doi.org/10.1093/gigascience/giz034>

Reviewer Reports on the First Revision:

Referees' comments:

Referee #1 (Remarks to the Author):

My queries have been dealt with and I recommend publication.

Couple of small things:

typo on line 138

Fig2A (also similar extended figure 7). The labels in the PCA group centroids are hard to read - perhaps change font colours.

Referee #2 (Remarks to the Author):

The revised paper has been significantly improved, both by addressing concerns raised in the original reports, and by adding a substantial body of additional data, notably from England (74 new samples from Dover-Buckland). These allow important new conclusions to be reached in addition to those originally presented. I am satisfied that the concerns raised in my previous report have been addressed.

A few minor corrections are needed:

Line 79: Frisian, not Friesian

Lines 250-51: Something is missing from the sentence in parentheses.

Lines 409 & 457: Merovingian is a sub-period relating to Frankish Gaul, so it is confusing to say 'Frankish and Merovingian'; better just to put 'Frankish'.

Author Rebuttals to First Revision:

Our responses are marked in blue.

Referee #1 (Remarks to the Author):

My queries have been dealt with and I recommend publication.

Couple of small things:

typo on line 138 -> Fixed

Fig2A (also similar extended figure 7). The labels in the PCA group centroids are hard to read - perhaps change font colours. -> We have changed the layout of this figure to conform to the required scales (figure width and font size).

Referee #2 (Remarks to the Author):

The revised paper has been significantly improved, both by addressing concerns raised in the original reports, and by adding a substantial body of additional data, notably from England (74 new samples from Dover-Buckland). These allow important new conclusions to be reached in addition to those originally presented. I am satisfied that the concerns raised in my previous report have been addressed.

A few minor corrections are needed:

Line 79: Frisian, not Friesian -> Fixed

Lines 250-51: Something is missing from the sentence in parentheses. -> Fixed

Lines 409 & 457: Merovingian is a sub-period relating to Frankish Gaul, so it is confusing to say 'Frankish and Merovingian'; better just to put 'Frankish'. -> Fixed